# 🦌 ReNO: Enhancing One-step Text-to-Image Models through Reward-based Noise Optimization

**Luca Eyring**[1,2,3,*]  **Shyamgopal Karthik**[1,2,3,4*]  **Karsten Roth**[2,3,4]
**Alexey Dosovitskiy**[5]  **Zeynep Akata**[1,2,3]

[1]Technical University of Munich  [2]Munich Center of Machine Learning
[3]Helmholtz Munich  [4]University of Tübingen & Tübingen AI Center  [5]Inceptive
luca.eyring@tum.de  shyamgopal.karthik@uni-tuebingen.de

## Abstract

Text-to-Image (T2I) models have made significant advancements in recent years, but they still struggle to accurately capture intricate details specified in complex compositional prompts. While fine-tuning T2I models with reward objectives has shown promise, it suffers from "reward hacking" and may not generalize well to unseen prompt distributions. In this work, we propose **Re**ward-based **N**oise **O**ptimization (**ReNO**), a novel approach that enhances T2I models at inference by optimizing the initial noise based on the signal from one or multiple human preference reward models. Remarkably, solving this optimization problem with gradient ascent for 50 iterations yields impressive results on four different one-step models across two competitive benchmarks, T2I-CompBench and GenEval. Within a computational budget of 20-50 seconds, ReNO-enhanced one-step models consistently surpass the performance of all current open-source Text-to-Image models. Extensive user studies demonstrate that our model is preferred nearly twice as often compared to the popular SDXL model and is on par with the proprietary Stable Diffusion 3 with 8B parameters. Moreover, given the same computational resources, a ReNO-optimized one-step model outperforms widely-used open-source models such as SDXL and PixArt-$\alpha$, highlighting the efficiency and effectiveness of ReNO in enhancing T2I model performance at inference time. Code is available at https://github.com/ExplainableML/ReNO.

## 1 Introduction

Advancements in Text-to-Image (T2I) models have been achieved in recent years, largely due to the availability of massive image-text datasets [26, 82, 83] and the development of denoising diffusion models [19, 36, 76, 84]. Despite these improvements, T2I models often struggle to accurately capture the intricate details specified in complex compositional prompts [3, 37]. Common challenges include incorrect text rendering, difficulties with attribute binding, generation of unlikely object combinations, and color leakage. While recent models have begun to address these issues by employing enhanced language encoders, larger diffusion models, and better data curation [6, 12, 13, 22], these approaches typically involve training larger models from scratch, making them inapplicable to existing models.

As a more efficient alternative, fine-tuning T2I models has gained significant attention. This approach can be tailored either toward specific preferences [32, 78, 105] or general human preferences. Inspired by the success of Reinforcement Learning from Human Feedback (RLHF) [16, 29] in Large Language

---

*equal contribution

38th Conference on Neural Information Processing Systems (NeurIPS 2024).

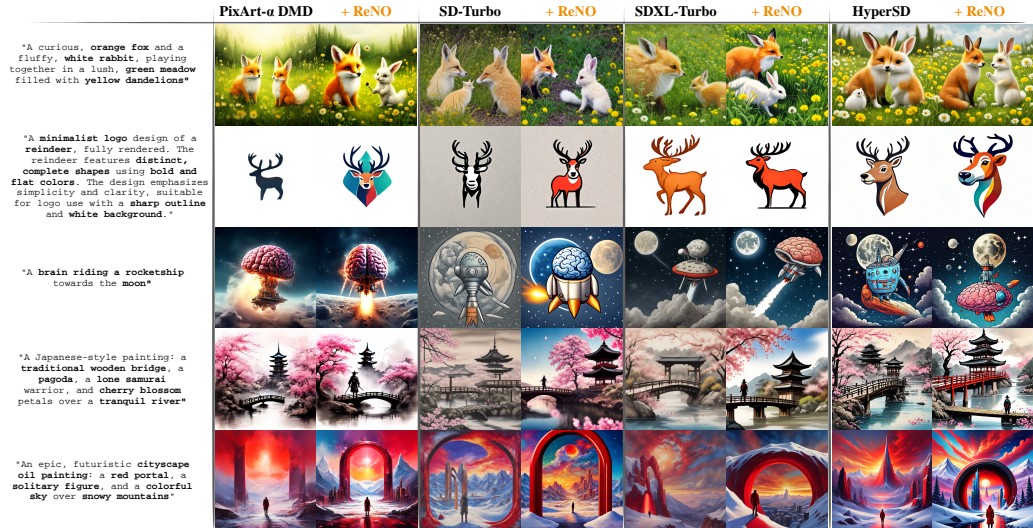

Figure 1: Qualitative results of four different one-step Text-to-Image models with and without ReNO over different prompts. The same initial random noise is used for the one-step generation and the initialization of ReNO. ReNO significantly improves upon the initially generated image with respect to both prompt faithfulness as well as aesthetic quality for all four models. Best viewed zoomed in.

Models (LLMs), several works [11, 18, 23, 74, 109] propose aligning T2I models by fine-tuning them on human-preferred prompt-image sets using RLHF-inspired techniques. Additionally, human preference reward models, such as PickScore [46], HPSv2 [97], and ImageReward [100], have gained popularity. These models are trained to output a score reflecting human preference for an image given a specific prompt, typically by measuring human preferences for various images generated from the same prompt. The scores predicted by these models have been utilized as evaluation metrics for the quality of generated images. Furthermore, Clark et al. [17], Li et al. [51], Prabhudesai et al. [72] directly fine-tune T2I models on these differentiable reward models to maximize the predicted reward of generated images. This approach is efficient due to the directly differentiable objective.

Fine-tuning T2I models with reward objectives has a major drawback of "reward hacking", which occurs when a reward model gives a high score to an undesirable image. Reward hacking points to deficiencies in existing reward models, highlighting gaps between the desired behavior and the behavior implicitly captured by the reward model, which is especially prone to appear when explicitly fine-tuning for a reward [17, 20, 51]. Additionally, these models are often fine-tuned on a small scale (e.g., <50 prompts in some cases [8, 23]) and thus may not generalize well to unseen prompt distributions with complex compositional structures. In this work, our aim is to enhance T2I models at *inference* for each unique generation, similar to the paradigm of test-time training for classification models [27, 88]. Fine-tuning diffusion models for every single prompt would both be computationally expensive (Dreambooth [78] takes 5 minutes on 1 A100), and susceptible to "reward-hacking".

We sidestep the challenge of fine-tuning the model's parameters and instead explore optimizing the initial random noise during *inference* without adapting any of the model's parameters. To obtain more optimal noise and a higher-quality generated image, we introduce Reward-based Noise Optimization (ReNO), where the initial noise is updated based on the signal from a reward model evaluated on the generated image. The main challenges in this approach are twofold. First, backpropagating the gradient through the denoising steps can lead to exploding/vanishing gradients, rendering the optimization process unstable. Our insight is that by employing a distilled *one-step* T2I model [12, 75, 81, 102], we circumvent the issue of exploding/vanishing gradients since backpropagation is performed through a single step. Second, naively optimizing the initial latent for an arbitrary objective can lead to collapse due to reward hacking. To mitigate this, we propose the use of a combination of reward objectives to not overfit to any single reward. Moreover, given a well-calibrated one-step T2I model with frozen parameters, the generated images should not exhibit reward hacking if the initial noise remains in the proximity of the initial noise distribution. Therefore, we propose an optimization scheme with limited steps, regularization of the noise to stay in-distribution, and gradient clipping.

In essence, ReNO involves optimizing the initial latent noise given an one-step T2I model (e.g., SD/SDXL-Turbo) and a reward model (e.g., ImageReward) for a limited number of iterations (10-50 steps). On the popular evaluation benchmarks T2I-Compbench and GenEval, our noise optimization strategy (ReNO) significantly improves performance, increasing scores by over 20% in some cases. This enhancement allows SD2.1-Turbo models to approach the performance of closed-source proprietary models such as DALL-E 3 [6] and SD3 [22]. We demonstrate that ReNO substantially improves the performance of four different one-step T2I models (e.g. Figure 1), both in terms of quantitative evaluation and extensive user studies, while only requiring 20-50 seconds to generate an image. Moreover, given the same computational budget, ReNO surpasses the performance of competing multi-step models, offering an attractive trade-off between performance and inference speed. ReNO not only motivates the development of more robust reward models but also provides a compelling benchmark for their evaluation. Finally, our results highlight the importance of the noise distribution in T2I models and encourage further research into understanding and adapting it.

## 2 Reward-based Noise Optimization (ReNO)

Despite the remarkable progress in Text-to-Image (T2I) generation, current state-of-the-art models still struggle to consistently produce visually satisfactory images that fully adhere to the input prompt. Recent studies have highlighted the significant impact of the initial noise vector $\varepsilon$ on the quality of the generated image [19, 85]. In fact, selecting and re-ranking images generated from a set of initial noises based on reward models has been shown to substantially improve performance [41, 46]. This observation naturally leads to the question of whether it is possible to identify an *optimal* noise vector that maximizes a given goodness measure for the generated image. In this section, we first provide an overview of one-step diffusion models, which serve as the foundation for our work. We then introduce our simple yet principled approach that enables practical noise optimization to enhance the performance of one-step T2I models based on human-preference reward models, addressing the challenge of generating high-quality images that align with the input prompt.

### 2.1 Background: One-Step Diffusion Models

T2I models aim to generate images $\mathbf{x}_0$ conditioned on a given textual prompt p. A generative model $G_\theta$ parameterized by $\theta$ takes as input a noise vector $\varepsilon \sim \mathcal{N}(0, \mathbf{I})$ and a prompt p, and outputs an image $G_\theta(\varepsilon, \text{p}) = \mathbf{x}_0$. The objective is to learn the parameters $\theta$, such that the generated image $\mathbf{x}_0$ aligns with the semantics of the prompt p. This is typically achieved by training the model on a large dataset of paired text and images. Recent models are based on a time-dependent formulation between a standard Gaussian distribution $\varepsilon \sim \mathcal{N}(0, \mathbf{I})$, and data $\mathbf{x}_0 \sim p_0(\mathbf{x})$. These models define a probability path between the initial noise distribution and the target data distribution, such that

$$\mathbf{x}_t = \alpha_t \mathbf{x}_0 + \sigma_t \varepsilon, \tag{1}$$

where $\alpha_t$ is a decreasing and $\sigma_t$ is an increasing function of $t$. Score-based diffusion [40, 45, 86] and flow matching [1, 54, 57] models share the observation that the process $\mathbf{x}_t$ can be sampled dynamically using a stochastic or ordinary differential equation (SDE or ODE). Consider the forward SDE that transforms data into noise as $t$ increases $d\mathbf{x}_t = \mathbf{u}(\mathbf{x}_t, t) \, dt + g(t) \, d\mathbf{w}_t$, where $\mathbf{u}_t(x_t, t)$ denotes the drift, $\mathbf{w}_t$ is a Wiener process and $g(t)$ represents the diffusion schedule. Then, the marginal probability distribution $p_t(\mathbf{x})$ of $\mathbf{x}_t$ in (1) coincides with the distribution of the probability flow ODE [45, 86], as well as the reverse-time SDE [2]

$$d\mathbf{x}_t = [\mathbf{u}(\mathbf{x}_t, t) - g(t)^2 \mathbf{s}(\mathbf{x}_t, t)] \, dt + g(t) \, d\bar{\mathbf{w}}_t, \tag{2}$$

where $\mathbf{s}(\mathbf{x}, t) = \nabla \log p_t(\mathbf{x})$ is the score function. By solving either the ODE or SDE backward in time from $\mathbf{x}_T = \varepsilon \sim \mathcal{N}(0, \mathbf{I})$, we can generate samples from $p_0(\mathbf{x})$. This relies on a good estimate of the parameterized score $\mathbf{s}_\theta(\mathbf{x}_t, t)$. The choice of functions $\alpha_t$ and $\sigma_t$ are defined implicitly based on the forward SDE [40, 45, 85, 87]. Furthermore, the process $\mathbf{x}_t$ is considered on an interval $[0, T]$ with $T$ sufficiently large such that $\mathbf{x}_T$ approximates the initial noise distribution $\mathcal{N}(0, \mathbf{I})$. Then, it has been shown that the score can be approximated efficiently based on, e.g. the denoising loss [36]

$$\mathcal{L}_s(\theta) = \mathbb{E}_{\mathbf{x}_0 \sim p(\mathbf{x}_0), \varepsilon \sim \mathcal{N}(0, \mathbf{I}), t \sim \mathcal{U}(0, T)}[\|\sigma_t \mathbf{s}_\theta(\mathbf{x}_t, t) + \varepsilon\|^2]. \tag{3}$$

During inference, these models simulate an ODE/SDE through discretization for a number of steps. This can be computationally expensive as the trained model must be evaluated sequentially.

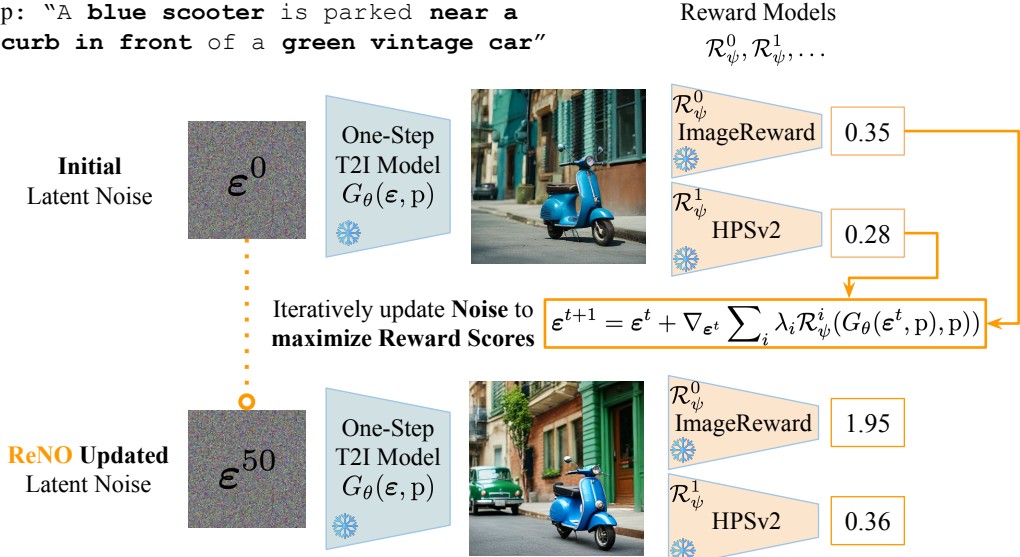

Figure 2: Overview of our proposed ReNO framework. Given reward models based on human preferences, we optimize the initial latent noise to maximize the reward scores (consisting HPSv2 [97], PickScore [46], ImageReward [100], and CLIP [73]) for the images generated by the one-step T2I model. Over 50 iterations, the quality of the images and the prompt faithfulness are improved.

**Distillation.**  As a means to reduce inference time, distillation techniques have recently gained traction with the intent to learn a student model that approximates the solution of the simulated differential equation with a trained teacher model given fewer inference steps, e.g. score distillation [71] penalizes the estimated score to the real data distribution. Furthermore, several methods have been proposed to distill models into *one-step* generators, which learn to approximate the full ODE or SDE in *one step*. Our work builds upon the following one-step T2I models which we refer to as $\tilde{G}_\theta$. Adversarial Diffusion Distillation (ADD) [81] combines score distillation with an adversarial loss and is employed to train SD-Turbo based on SD 2.1 [76] as a teacher and SDXL-Turbo [81] based on SDXL [69]. Diffusion Matching Distillation (DMD) [102] additionally leverages a distributional loss based on an approximated KL divergence and is applied for PixArt-$\alpha$ DMD [12, 13]. Lastly, Trajectory Segmented Consistency Distillation (TSCD) [75] introduces a progressive segment-wise consistency distillation [44, 87] loss to train HyperSDXL [75] with reward fine-tuning. All these models are trained in latent space such that during inference, an image is generated by first generating a sample in latent space and then decoding it $G_\theta(\varepsilon, p) = \mathcal{D}(\tilde{G}_\theta(\varepsilon, p))$ with a pre-trained decoder $\mathcal{D}$.

## 2.2   Initial Noise Optimization

Given a Text-to-Image generative model $G_\theta(\varepsilon, p)$ that generates images based on a noise $\varepsilon$ and a prompt p, we defined the following optimization problem following previous work [5, 43, 80, 91] with the objective of optimizing the noise $\varepsilon$ based on a criterion function $\mathcal{C} : \mathbb{R}^{H \times W \times c} \to \mathbb{R}$ evaluated on the generated image

$$\varepsilon^\star = \arg\max_{\varepsilon} \mathcal{C}(G_\theta(\varepsilon, p)). \tag{4}$$

Then, given a differentiable $\mathcal{C}$, (4) can be solved through iterative optimization via standard gradient ascent techniques. However, backpropagating through $\mathcal{C}(G_\theta(\varepsilon, p))$ is non-trivial as current Text-to-Image models are based on the simulation of ODEs or SDEs (Section 2.1). Several methods have been proposed to enable backpropagation through time-dependent generative models [14, 17, 60, 91], based on e.g., the adjoint method [70]. Our method, in contrast, leverages the crucial observation that selecting a one-step model as $G_\theta$ enables efficient backpropagation through (4). Although this realization may initially appear trivial, it proves to be a fundamental step in facilitating practical noise optimization in Text-to-Image models. Current methods require between 10 [91] and 40 [5] minutes to optimize noise and thus, to generate a single image. Our approach achieves image generation, including noise optimization, in 20-50 seconds, making it suitable for practical applications.

**Noise regularization.** One important consideration, is that it is desirable for $\varepsilon$ to stay within the proximity of the initial noise distribution $\mathcal{N}(0, \mathbf{I})$ as otherwise $G_\theta$ might provide unwanted generations. This can be realized by including a regularization term inside of $\mathcal{C}$. Samuel et al. [79] propose instead of directly optimizing the likelihood of $p_T(\varepsilon)$, to instead consider the likelihood of the norm of the noise $r = ||\varepsilon||$, which is distributed according to a $\chi^d$ distribution $p(r)$. Thus, following Ben-Hamu et al. [5], Samuel et al. [80] we maximize the log-likelihood of the norm of a noise sample $K(\varepsilon) = (d-1)\log(||\varepsilon||) - ||\varepsilon||^2/2$. In our framework, this corresponds to employing a regularized criterion function given by $\mathcal{C}(\mathbf{x}_0, \varepsilon) = \tilde{\mathcal{C}}(\mathbf{x}_0) + K(\varepsilon)$, which can be plugged into (4).

In Figure 3, we provide an illustrative example where we chose the criterion to maximize a selected color channel $c$ of the generated image while minimizing the other two $\bar{c}_1, \bar{c}_2$

$$\tilde{\mathcal{C}}(\mathbf{x}_0) = \sum\nolimits_{i,j} \mathbf{x}_0^{i,j,c} - \mathbf{x}_0^{i,j,\bar{c}_1} - \mathbf{x}_0^{i,j,\bar{c}_2}, \quad (5)$$

where $\mathbf{x}_0^{i,j,c}$ denotes the channel $c$ of the pixel at $(i,j)$. Note that due to the calibration of the trained model and the noise staying in-distribution, the noise does not collapse to the optimal $\varepsilon^\star$, which would result in the generation of a fully blue or red image. Also, the optimization first adapts the color of the car and then starts changing the background. Here, 10 optimization steps provide satisfactory results illustrating the efficacy of the proposed one-step noise optimization framework.

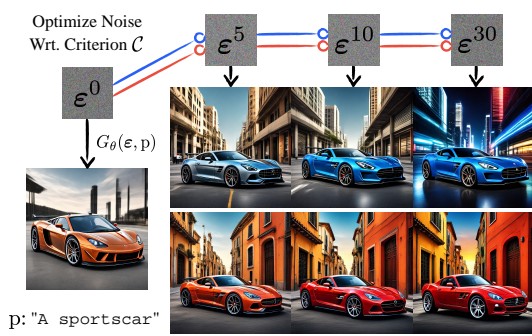

Figure 3: Initial noise optimization for one-step $G_\theta$ HyperSDXL with two color channel criterions (5).

## 2.3 Human Preference Reward Models and Our Reward Criterion

Inspired by the success of Reinforcement Learning From Human Feedback [16, 29] in aligning LLMs with human preferences, similar methods have been explored for T2I generation. The underlying idea is to train a model $\mathcal{R}_\psi$ that takes in an input along with the generated output (in this case a prompt and the corresponding image) and provides a score for the "goodness" of the generated output. Notable open-source human preference reward models for T2I include ImageReward [100] based on BLIP [49] and human preferences collected for the DiffusionDB dataset, PickScore [46], and HPSv2 [97] both based on a CLIP [73] ViT-H/14 backbone. These reward models provide a quantitative measure of the image's quality and relevance to the prompt through a prediction by a differentiable neural network. Thus, they have not only been employed for the evaluation of T2I models but also to fine-tune them [17, 18, 100] as a means of achieving higher reward scores. Lastly, CLIPScore [35] has also been leveraged to measure the prompt alignment of a generated image.

To *generally* enhance the performance of Text-to-Image models *without* any fine-tuning, we propose to leverage a **Re**ward-based criterion function $\mathcal{C}$ for **N**oise **O**ptimization (**ReNO**). Specifically, we propose to use a weighted combination of a number $n$ of pre-trained reward models $\mathcal{R}_\psi^0, \dots \mathcal{R}_\psi^n$ as the criterion function

$$\tilde{\mathcal{C}}(\mathbf{x}_0, \mathrm{p}) = \sum\nolimits_i^n \lambda_i \mathcal{R}_\psi^i(\mathbf{x}_0, \mathrm{p}), \quad (6)$$

where $\lambda_i$ denotes the weighting for reward model $\mathcal{R}_\psi^i$. Employing a combination of reward models can help prevent "reward-hacking" and allow capturing various aspects of image quality and prompt adherence, as different reward models are trained on different prompt and preference sets. This not only effectively combines the strengths of multiple reward models, but also helps mitigate their weaknesses. **ReNO** then boils down to iteratively solving (4) with gradient ascent

$$\varepsilon^{t+1} = \varepsilon^t + \eta \nabla_{\varepsilon^t} [K(\varepsilon^t) + \sum\nolimits_i^n \lambda_i \mathcal{R}_\psi^i(G_\theta(\varepsilon^t, \mathrm{p}), \mathrm{p})], \quad (7)$$

where $\eta$ is the learning rate. Similar to the color example in Figure 3, it is actually not desirable to find the optimal $\varepsilon^\star$ as we want to prevent adversarial samples that exploit the reward models. We find that already a few optimization steps (<50) of **ReNO** lead to significant improvements in both prompt following and visual aesthetics, striking a good balance between reward optimization and the prevention of reward hacking. Due to the efficacy of the proposed framework, generating one image, including noise optimization, takes between 20-50 seconds, depending on the model and image size, enabling its practical use. We provide a sketch of **ReNO** in Figure 2 and full details in Algorithm 1.

# 3 Related Work

**Initial Noise Optimization.** The initial noise optimization framework was first introduced in DOODL [91] for improved guidance in Text-to-Image models. Subsequently, it was leveraged by Karunratanakul et al. [43] for 3D universal motion priors, for rare-concept generation [61, 79, 80] and enhancing image quality [31, 89] in text-to-image models, music generation [62, 63], and by D-Flow [5] for solving inverse problems in various settings. While these methods mainly focus on controlling the generated sample for specific applications, our proposed method is designed to *generally* improve Text-to-Image models without the need for additional techniques to mitigate exploding or vanishing gradients on the optimization process. Most related to our work is DOODL [91], which also proposes to improve the textual alignment of text-to-image models using a CLIP-score-based criterion function, which we similarly employ in our method. These existing methods, however, take 10 (DOODL) to 40 (D-Flow) minutes to generate a single image due to their application on time-dependent generative models with a large number of denoising steps. To mitigate this, Samuel et al. [80] propose a bootstrap-based method to increase the efficiency of generating a batch of images. However, this method is limited to settings where the goal is to generate samples including a concept jointly represented by a set of input images.

**Reward Optimization for Text-to-Image Models.** Reward models [46, 47, 97, 98, 100] were first introduced to mimic human preferences given an input prompt and generated images. There have been several attempts at incorporating these signals to enhance text-to-image generation. One notable direction is the idea of using reinforcement learning based algorithms to fine-tune text-to-image models to better align with these rewards either with an explicit reward model [8, 11, 18, 23, 30, 109] or by bypassing it entirely with Direct Preference Optimization [50, 74, 92, 101]. However, this can be expensive, requiring thousands of queries to generalize, and therefore a lot of work has explored directly fine-tuning diffusion models [17, 51, 72] using differentiable rewards [39, 46, 47, 98, 100]. Additionally, there has also been works exploring the concept of using reward models to perform classifier-guidance [4, 34] as well as using rewards to distill diffusion models into fewer steps [48, 75]. Differently from these works, we focus on adapting a diffusion model during inference by purely optimizing the initial latent noise using a differentiable objective.

# 4 Experiments

**Experimental Setup.** We evaluate the effectiveness of our proposed method, ReNO, using four open-source one-step image generation models: SD-Turbo, SDXL-Turbo, PixArt-$\alpha$ DMD, FLUX-schnell and HyperSDXL. HyperSDXL generates images of size $1024 \times 1024$ while the others generate $512 \times 512$. To assess the performance across diverse scenarios, we consider three challenging tasks. First, we evaluate on T2I-CompBench [37], which comprises 6000 compositional prompts spanning six categories, using a VQA, object detection, and image-text matching scores. Second, we employ GenEval [28], consisting of 552 object-focused prompts, measuring the quality of the generated images using a pre-trained object detector. Finally, we utilize Parti-Prompts [103], a collection of more than 1600 complex prompts, and assess the generated images using both reward-based metrics and extensive human evaluation. Throughout all experiments, we optimize Equation (7) for 50 steps using gradient ascent with Nesterov momentum and gradient norm clipping for stability. Lastly, we select the image with the highest reward score from the optimization trajectory for evaluation.

## 4.1 Effect of Reward Models

We analyze the effect of various reward models in Table 1. We see that optimizing ImageReward or CLIPScore alone improves the text-image faithfulness (i.e., attribute binding from T2I-Compbench). However, this comes at the cost of decreased aesthetic score. PickScore and HPSv2 improve the image quality, however the gains in faithfulness are modest. Combining all the rewards leads to having strong improvements in faithfulness, while

Table 1: SD-Turbo evaluated on the attribute binding categories of T2I-CompBench and the LAION aesthetic score predictor [83] for different reward models.

| Reward | Attribute Binding | | | Aesthetic ↑ |
|---|---|---|---|---|
| | Color ↑ | Shape ↑ | Texture ↑ | |
| SD-Turbo | 0.5513 | 0.4448 | 0.5690 | 5.647 |
| + CLIPScore | 0.6625 | 0.5501 | 0.6621 | 5.475 |
| + HPSv2 | 0.6443 | 0.5451 | 0.6859 | 5.752 |
| + ImageReward | 0.7720 | 0.6104 | 0.7334 | 5.611 |
| + PickScore | 0.6341 | 0.5069 | 0.6242 | 5.711 |
| + All | 0.7830 | 0.6244 | 0.7466 | 5.704 |

Table 2: **Quantitative Results on T2I-CompBench**. ReNO combined with (1) PixArt-$\alpha$ DMD [12, 13, 102], (2) SD-Turbo [81], (3) SDXL-Turbo [81], (4) HyperSD [75] demonstrates superior compositional generation ability in both attribute binding, object relationships, and complex compositions. The best value is bolded, and the second-best value is underlined. Multi-step results taken from [13, 22].

| Model | Attribute Binding | | | Object Relationship | | Complex↑ |
| --- | --- | --- | --- | --- | --- | --- |
| | Color ↑ | Shape↑ | Texture↑ | Spatial↑ | Non-Spatial↑ | |
| SD v1.4 | 0.38 | 0.36 | 0.42 | 0.12 | 0.31 | 0.31 |
| SD v2.1 | 0.51 | 0.42 | 0.49 | 0.13 | 0.31 | 0.34 |
| SDXL | 0.64 | 0.54 | 0.56 | 0.20 | 0.31 | 0.41 |
| PixArt-$\alpha$ | 0.69 | 0.56 | 0.70 | 0.21 | **0.32** | 0.41 |
| DALL-E 2 | 0.57 | 0.55 | 0.64 | 0.13 | 0.30 | 0.37 |
| DALL-E 3 | **0.81** | **0.68** | **0.81** | - | - | - |
| (1) PixArt-$\alpha$ DMD | 0.38 | 0.34 | 0.47 | 0.19 | 0.30 | 0.36 |
| **(1) + ReNO (Ours)** | 0.64 | 0.57 | 0.72 | 0.25 | 0.31 | 0.46 |
| (2) SD-Turbo | 0.55 | 0.44 | 0.57 | 0.17 | 0.31 | 0.41 |
| **(2) + ReNO (Ours)** | 0.78 | 0.62 | 0.75 | 0.22 | **0.32** | **0.48** |
| (3) SDXL-Turbo | 0.61 | 0.44 | 0.60 | 0.24 | 0.31 | 0.43 |
| **(3) + ReNO (Ours)** | 0.78 | 0.60 | 0.74 | **0.26** | 0.31 | 0.47 |
| (4) HyperSDXL | 0.65 | 0.50 | 0.65 | 0.25 | 0.31 | 0.46 |
| **(4) + ReNO (Ours)** | 0.79 | 0.63 | 0.77 | **0.26** | 0.31 | **0.48** |

Table 3: **Quantitative Results on GenEval**. ReNO combined with (1) PixArt-$\alpha$ DMD [12, 13, 102], (2) SD-Turbo [81], (3) SDXL-Turbo [81], (4) HyperSDXL [75] improves results across all categories. The best value is bolded, and the second-best value is underlined. Multi-step results taken from [22].

| Model | Mean ↑ | Single↑ | Two↑ | Counting↑ | Colors↑ | Position↑ | Color Attribution↑ |
| --- | --- | --- | --- | --- | --- | --- | --- |
| SD v2.1 | 0.50 | 0.98 | 0.51 | 0.44 | 0.85 | 0.07 | 0.17 |
| SDXL | 0.55 | 0.98 | 0.74 | 0.39 | 0.85 | 0.15 | 0.23 |
| IF-XL | 0.61 | 0.97 | 0.74 | 0.66 | 0.81 | 0.13 | 0.35 |
| PixArt-$\alpha$ | 0.48 | 0.98 | 0.50 | 0.44 | 0.80 | 0.08 | 0.07 |
| DALL-E 2 | 0.52 | 0.94 | 0.66 | 0.49 | 0.77 | 0.10 | 0.19 |
| DALL-E 3 | 0.67 | 0.96 | 0.87 | 0.47 | 0.83 | **0.43** | 0.45 |
| SD3 (8B) | 0.68 | 0.98 | 0.84 | 0.66 | 0.74 | 0.40 | 0.43 |
| (1) PixArt-$\alpha$ DMD | 0.45 | 0.95 | 0.38 | 0.46 | 0.76 | 0.05 | 0.09 |
| **(1) + ReNO (Ours)** | 0.59 | 0.98 | 0.72 | 0.58 | 0.85 | 0.15 | 0.27 |
| (2) SD-Turbo | 0.49 | 0.99 | 0.51 | 0.38 | 0.85 | 0.07 | 0.14 |
| **(2) + ReNO (Ours)** | 0.62 | **1.00** | 0.82 | 0.60 | 0.88 | 0.12 | 0.33 |
| (3) SDXL-Turbo | 0.54 | **1.00** | 0.66 | 0.45 | 0.84 | 0.09 | 0.20 |
| **(3) + ReNO (Ours)** | 0.65 | **1.00** | 0.84 | 0.68 | 0.90 | 0.13 | 0.35 |
| (4) HyperSDXL | 0.56 | **1.00** | 0.76 | 0.43 | 0.87 | 0.10 | 0.21 |
| **(4) + ReNO (Ours)** | 0.65 | **1.00** | **0.90** | 0.56 | **0.91** | 0.17 | 0.33 |
| FLUX-schnell | 0.64 | 0.98 | 0.80 | 0.64 | 0.78 | 0.18 | 0.43 |
| FLUX-schnell **+ ReNO (Ours)** | **0.72** | 0.99 | **0.90** | 0.79 | 0.87 | 0.21 | **0.56** |
| FLUX-dev | 0.68 | 0.99 | 0.85 | 0.74 | 0.79 | 0.21 | 0.48 |

also increasing the image quality. Thus, we employ ReNO with all four reward models. We report further details in Appendix C, including the performance of all combinations of reward models.

## 4.2 Quantitative Results

Table 2 presents the quantitative results of ReNO on T2I-Compbench. Most notably, we observe that for both Pixart-$\alpha$ DMD and SD-Turbo, there are improvements of over 20% in the Color, Shape, and Texture Categories. For instance, on Color SD-Turbo improves from 55% to 78%, which is only slightly below DALL-E 3. Similar improvements can also be seen for SDXL-Turbo and HyperSDXL models where performance increases by over 10 percentage points in these categories. Even outside this, there are significant boosts in the Spatial, Non-Spatial, and Complex categories, highlighting

both the efficacy of the noise optimization framework, as well as the utility of human preference models for improving T2I generation at inference. Similar trends can also be noticed for GenEval in Table 3, where applying our noise optimization framework helps improve the performance of various one-step diffusion models. For instance, SD-Turbo improves its mean score from 0.49 to 0.62. Notably, our strongest model, HyperSDXL + ReNO, comes very close to the proprietary DALL-E 3 and SD3, i.e., beating DALL-E 3 on 4/6 categories in GenEval. In the case of FLUX-schnell, ReNO improves the performance (0.72) to even surpass that of the base FLUX-dev model (0.68). Most notably, this is the strongest open-source results reported on the GenEval benchmark. In both of these benchmarks, our noise optimization framework improves results for all the models in all the categories. It is also important to note that both T2I-Compbench and GenEval use a variety of methods unrelated to human preference rewards, such as VQA models and object detectors, to detect different objects in the generated images. We report further quantitative results including comparisons to other test-time-based methods in Appendix B. Additionally, these quantitative results are supported by the qualitative results reported in Figure 1 and Appendix A. Lastly, we report full details for the conducted FLUX-schnell experiments in Appendix E.8.

### 4.3 User Study Results

To further validate ReNO we perform a user study on the commonly used Parti-Prompts [103] with Amazon Mechanical Turk (AMT). Parti-Prompts generally includes longer complex prompts that test artistic generation capabilities as opposed to T2I-Compbench and GenEval, which purely focus on faithfulness. We conducted user studies with ReNO applied to SD-Turbo for $512 \times 512$ and HyperSDXL for $1024 \times 1024$ generation. We compare SD-Turbo + ReNO against SD-Turbo, SDXL-Turbo, SD2.1

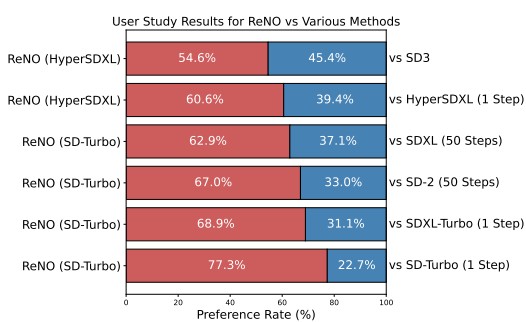

Figure 4: User Study Results for ReNO

(50 Steps), and SDXL-Base (50 Steps). The results in Figure 4 confirm our findings in the quantitative evaluation. SD-Turbo + ReNO has an above 60% win rate against all benchmarked models reaching up to 77% against the SD-Turbo base. To contextualize these results, SD3 [22] conducts a similar user study on Parti-Prompts and reports a 70% win rate against SDXL (50 steps). Our strongest base model, HyperSDXL, already beats SDXL (50 steps) [75] without ReNO. Thus, we compare it with and without ReNO as well as against the proprietary SD3 (8B) [22]. Again, HyperSDXL + ReNO achieves an above 60% win rate, and notably, it also narrowly beats SD3 with 54%. This confirms our finding in ReNO, which substantially improves overall generative quality, pushing results at least close to the ones of even current state-of-the-art proprietary models. Lastly, we note that user studies on AMT can potentially be noisy and, therefore, view the results holistically along with quantitative evaluation. We provide a detailed breakdown of the preference for image quality and faithfulness, as well as full details of the user study in Appendix D.

### 4.4 Computational Cost of ReNO

The primary concern of our proposed method is the increased inference cost since existing methods (e.g. DOODL, D-Flow) are impractical for regular T2I generation usage. However, we circumvent this issue through our restriction to one-step models and 50 optimization steps, which makes ReNO run in 20-50 seconds. To analyze the performance of ReNO with respect to the number of optimization steps we evaluate its performance over a set of reference points. We report results on the attribute binding part of T2I-CompBench for SD-Turbo + ReNO in

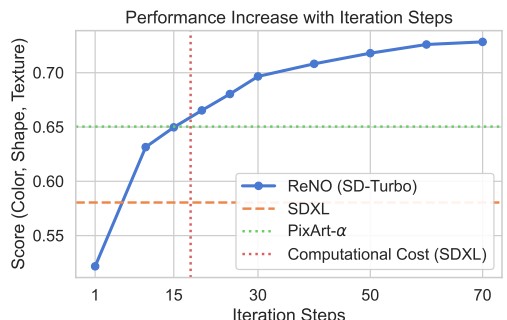

Figure 5: Attribute binding results on T2I-CompBench with varying number of iterations.

Figure 5 and visually corroborate these results with Figure 6. Note that even when restricted to the same compute budget as SDXL (50 steps, ~7sec), SD-Turbo + ReNO significantly outperforms it while in this comparison PixArt-$\alpha$ (20 steps, ~7sec) lies shortly below the Pareto-frontier of ReNO.

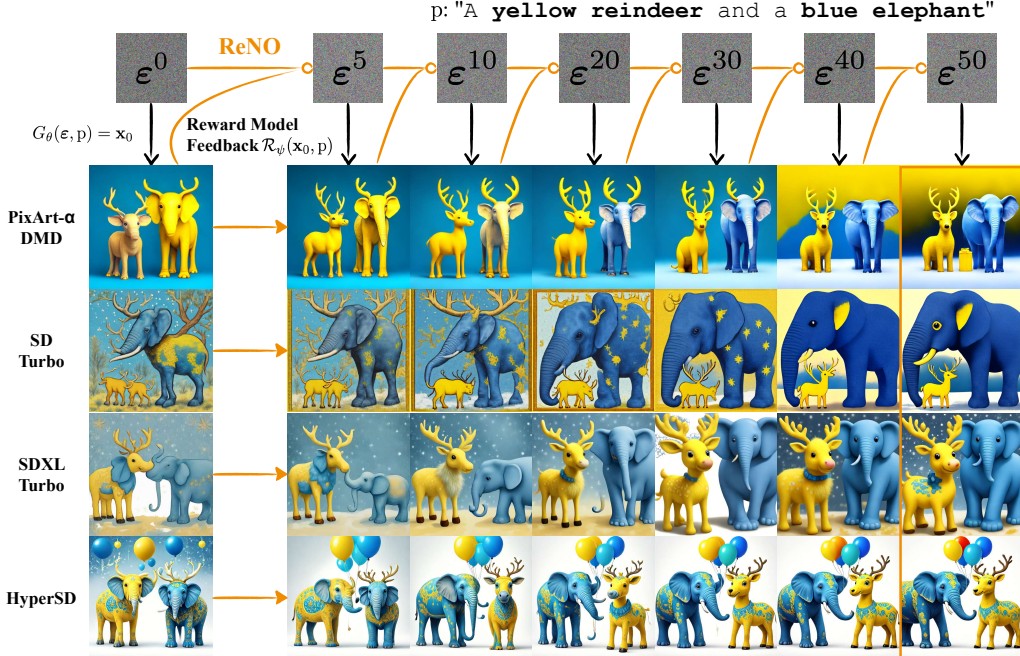

Figure 6: The initial images are generated with four different one-step models $G_\theta$ given the prompt p "A yellow reindeer and a blue elephant" and randomly initialized noise $\varepsilon^0$. Each column shows the result of optimizing the noise latent $\varepsilon^t$ for $t$ steps with respect to our reward-based criterion.

## 4.5   Effect of ReNO on the Diversity of Generated Images

To investigate the effect of noise optimization on output diversity, we evaluate images generated across 50 different random seeds for 110 prompts from Parti-Prompts. Specifically, we generate a batch of images and use LPIPS [106] and DINO [9, 64] scores to compute the average similarity of the generated batch, where a lower similarity score corresponds to higher diversity. As shown in Table 4, one-step models (SD-Turbo, SDXL-Turbo) exhibit lower diversity compared to their multi-step counterparts (SD2.1, SDXL), likely due to adver-

Table 4: We measure the average LPIPS and DINO similarity scores over images generated for 50 different seeds for 100 prompts from Parti-Promtps.

|  | LPIPS ↓ | DINO ↓ |
|---|---|---|
| SD-Turbo | $0.382 \pm 0.043$ | $0.770 \pm 0.101$ |
| **SD-Turbo + ReNO** | $\underline{0.246} \pm 0.046$ | $\underline{0.712} \pm 0.132$ |
| SD2.1 (50-step) | $\mathbf{0.243} \pm 0.049$ | $\mathbf{0.623} \pm 0.150$ |
| SDXL-Turbo | $0.391 \pm 0.044$ | $0.835 \pm 0.073$ |
| **SDXL-Turbo + ReNO** | $\mathbf{0.291} \pm 0.041$ | $\underline{0.763} \pm 0.116$ |
| SDXL (50-step) | $\underline{0.351} \pm 0.042$ | $\mathbf{0.700} \pm 0.128$ |

sarial training. However, applying ReNO not only maintains but actually *increases* diversity. For both SD-Turbo and SDXL-Turbo, ReNO achieves diversity levels approaching their respective multi-step base models highlighting an unexpected benefit of noise optimization increasing diversity. Figure 10 illustrates these improvements qualitatively. We hypothesize that the reason for this increased diversity is that ReNO adds structure to the noise, thus optimizes it away from the zero mean of the noise distribution and creating more diverse noises compared to sampling from the standard Gaussian.

## 4.6   Comparison to Multi-Step Noise Optimization

We benchmark ReNO against DOODL [91], which performs noise optimization using the 50-step SD2.1 model. Due to DOODL's computational demands, we evaluate on the first 50 prompts from T2I-CompBench. Despite using the same CLIPScore objective, ReNO achieves four times larger improvements in the optimized criterion while requiring 75% less GPU memory and running 100x faster, highlighting the effectiveness of our one-step approach. Moreover, ReNO's multi-reward objective leads to substantially larger gains in attribute binding accuracy (21.0-28.9%) compared to solely using CLIPScore, reiterating the efficacy of ReNO's optimization objective.

Table 5: Performance comparison of ReNO and DOODL over the first 50 prompts of each of the Attribute Binding categories in T2I-CompBench. We report scores from default T2I-Compbench evaluation using BLIP-VQA as well as the optimized CLIPScore before and after optimization.

| Model | Attribute Binding (Change) | | | CLIPScore ↑ | sec/iter (total) | VRAM |
|---|---|---|---|---|---|---|
| | Color ↑ | Shape↑ | Texture↑ | | | |
| SD2.1 | 33.4 | 52.4 | 63.4 | 0.261 | - | - |
| SD2.1 + DOODL (CLIP) | 38.5 (+5.1) | 51.6 (-0.8) | 64.6 (+1.2) | 0.289 (+0.03) | 24s (20min) | 40GB |
| SD-Turbo | 60.4 | 48.5 | 61.8 | 0.362 | - | - |
| SD-Turbo + ReNO (only CLIP) | 70.1 (+9.7) | 66.9 (+18.4) | 79.6 (+18.2) | 0.483 (**+0.12**) | 0.2 (10s) | 10GB |
| SD-Turbo + ReNO (all) | 82.1 (**+21.7**) | 77.4 (**+28.9**) | 82.8 (**+21.0**) | 0.437 (+0.08) | 0.4 (20s) | 15GB |

## 4.7 Limitations

An interesting observation in our experiments is that despite using different image generation models of varying architectures and sizes, they broadly converge to similar performance on both T2I-Compbench and GenEval. In addition to the limitations of the generative models, we hypothesize that this could be due to the limitations of the reward models themselves, given their limited compositional reasoning abilities [104]. Stronger reward models [53, 107] and preference data [15, 33, 42, 99, 108] would be crucial in enhancing results further.

Secondly, not only the runtime but also the amount of needed GPU VRAM is significantly higher when using ReNO. We reduce it by leveraging fp16 quantization and the `pytorch` [67] memory reduction technique introduced in Bhatia and Dangel [7], which for ReNO lowers the VRAM by another ~15%. Then, all of the models can be optimized on a single A100 GPU in 20-50

Table 6: Computational cost comparison of ReNO optimizing four reward models on an A100 GPU.

| Method | sec/iter (total) | VRAM | #params | Img size |
|---|---|---|---|---|
| SD-Turbo | 0.4s (20s) | 15GB | 860M | $512 \times 512$ |
| SDXL-Turbo | 0.6s (30s) | 21GB | 2.6B | $512 \times 512$ |
| PixArt-$\alpha$ DMD | 0.5s (25s) | 25GB | 600M | $512 \times 512$ |
| FLUX-schnell | 0.6s (30s) | 50GB | 12B | $512 \times 512$ |
| HyperSDXL | 1.0s (50s) | 39GB | 2.6B | $1024 \times 1024$ |

seconds, and e.g., SD-Turbo requires only 15GB VRAM for the entire optimization process. Note, however, that the amount of VRAM also scales with the size of the generated image. Thus, HyperS-DXL needs 39GB of VRAM. We provide a summary of the computational cost of ReNO in Table 6, which lays out ReNO's main limitation. Finally, current T2I models struggle with generating humans, rendering text, and also modeling complex compositional relations. While our work attempts to alleviate these issues and provides a flexible framework for further improvements, future work is required to resolve these issues.

## 5 Conclusion

We introduce ReNO, a test-time optimization strategy for enhancing text-to-image generation without any fine-tuning. Not only do we achieve the strongest results among all open-source models on T2I-Compbench and GenEval, but images from ReNO on a single-step SD-Turbo have over a 60% win rate against a 50-step SDXL model and is competitive with the 8B parameter SD3 model on user studies. We also demonstrate that ReNO outperforms SDXL even when restricted to the same computational budget, highlighting the benefits of ReNO for practical use cases. The performance gains from ReNO underscore the importance of developing even better and more robust reward models and, moreover, establish a valuable benchmark for assessing their effectiveness. Furthermore, the substantial impact of optimizing the initial noise distribution motivates further research into understanding, manipulating, and controlling this crucial aspect of generative models.

## Acknowledgements

This work was supported by BMBF FKZ: 01IS18039A, by the ERC (853489 - DEXIM), by EXC number 2064/1 – project number 390727645. Shyamgopal Karthik and Karsten Roth thank the International Max Planck Research School for Intelligent Systems (IMPRS-IS) for support. Luca Eyring and Karsten Roth would also like to thank the European Laboratory for Learning and Intelligent Systems (ELLIS) PhD program for support.

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

# Appendix

The Appendix is organized as follows:

# A   Further Qualitative Results

Here, we report further qualitative results. We separate them into $512 \times 512$ and $1024 \times 1024$ generated images. First, in Figure 7, we show examples of ReNO applied to SD-Turbo, SDXL-Turbo, and PixArt-$\alpha$ DMD. Then, in Figures 8 and 9, we report HyperSDXL + ReNO against competing methods. Broadly, we see that ReNO not only fixes the artifacts occurring in one-step models but also improves compositional understanding (color attribution, spatial reasoning) as well as the quality of generated faces and pushes current one-step models to be broadly on par with proprietary models in these settings. Lastly, we report qualitative results for the effect of ReNo on the diversity of generated images in Figure 10.

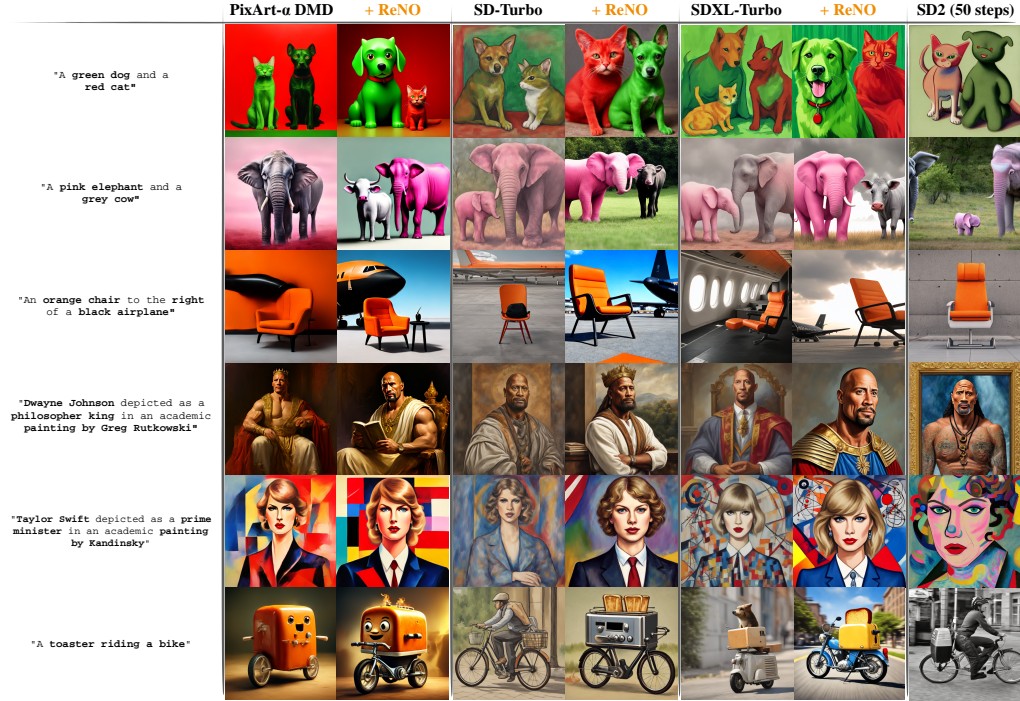

Figure 7: Comparison of images generated with and without ReNO at $512 \times 512$ resolution across various one-step models and SD v2.1. The noise used to generate the initial image is the same one that is used to initialize ReNO.

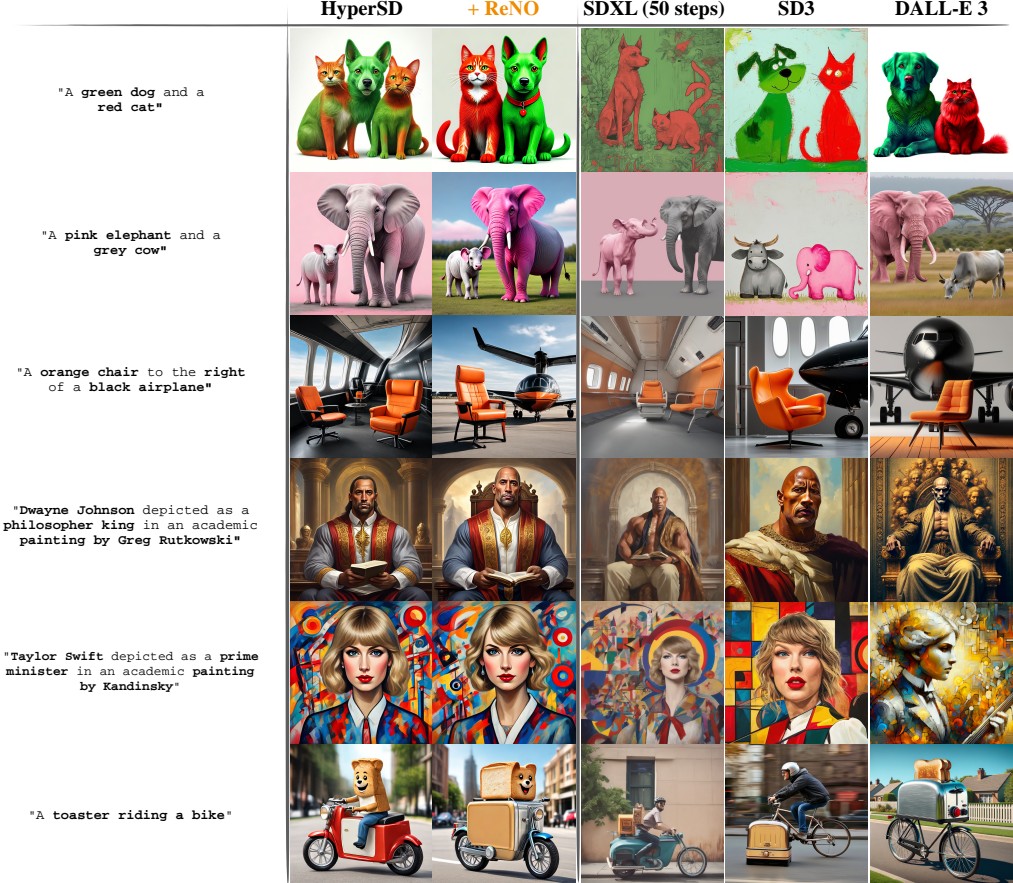

Figure 8: Images generated with and without ReNO using HyperSDXL at $1024 \times 1024$ resolution compared to competing T2I models SDXL, SD3, and DALL-E 3. ReNO helps to fix artifacts and generates images of comparable quality to even closed-source models. The noise used to generate the initial image is the same one that is used to initialize ReNO.

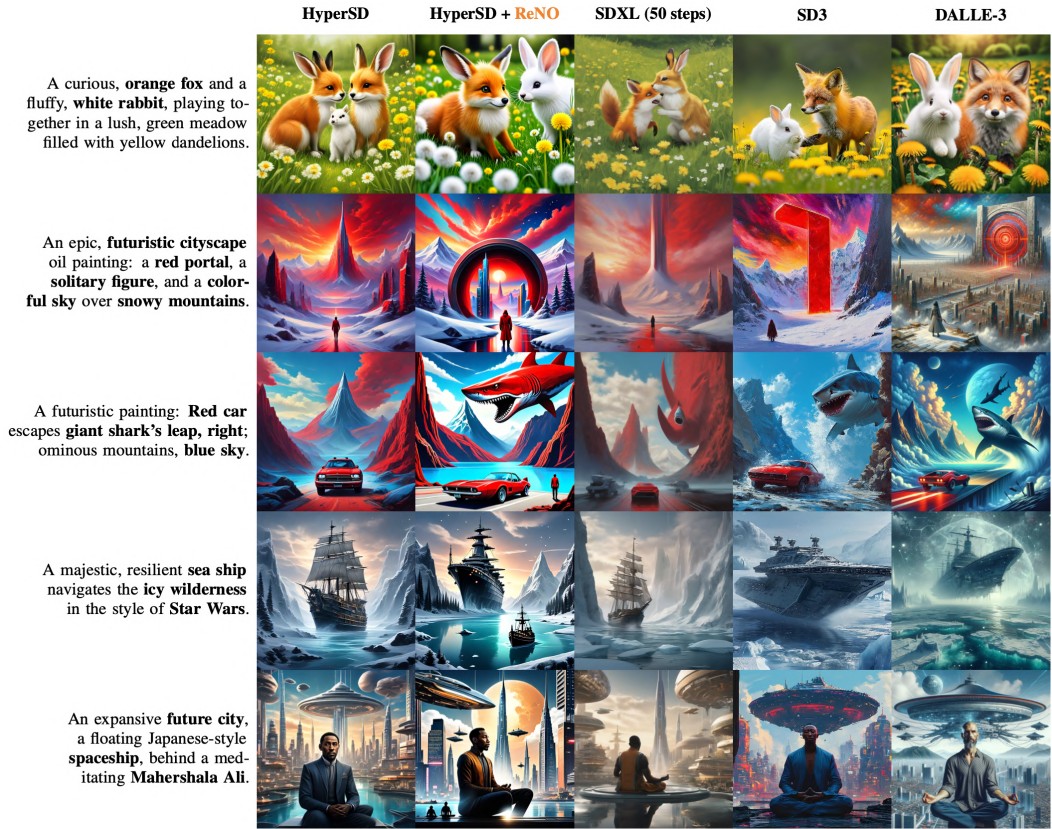

Figure 9: Comparison of generated images from different models (HyperSD, HyperSD + ReNO, SDXL (50 steps), SD3, DALLE-3) for various prompts. Each row corresponds to a specific prompt, and each column represents a different model.

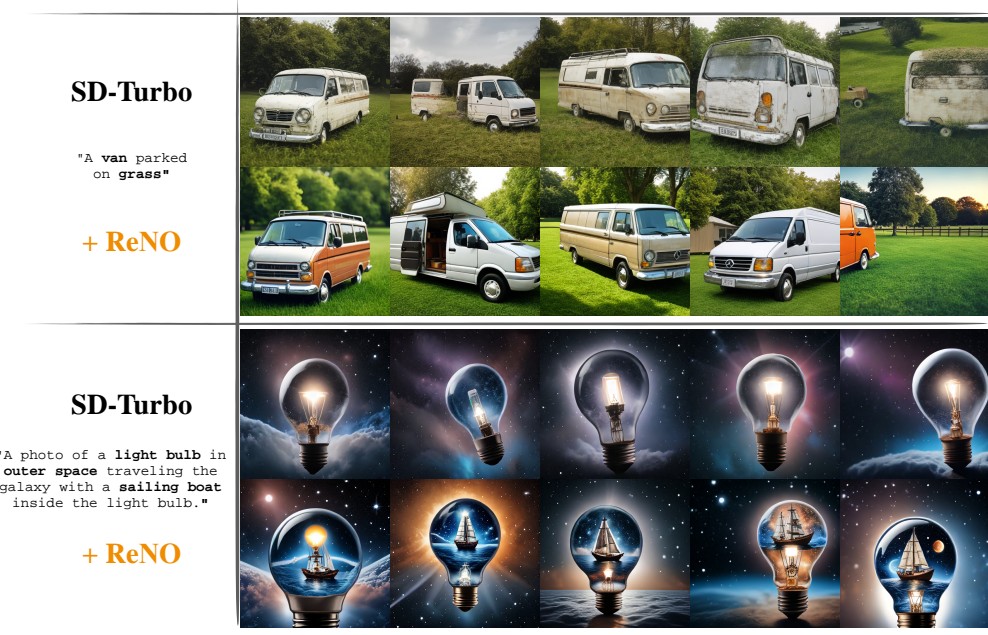

Figure 10: Non-cherry-picked results for SD-Turbo with and without **ReNO** for two different prompts over the first 5 seeds. ReNO increases the diversity of generated images w.r.t. content and layout.

# B  Further Quantitative Results

## B.1  Comparison to Direct Preference Optimization

Direct Preference Optimization has recently been applied in the context of Diffusion models [8, 50, 101]. Here, we compare against an SDXL model that has been preference-tuned on a dataset of over 800k preferences in Table 7. We see that while DPO improves both attribute binding and the aesthetic score of the generated images, it underperforms the SDXL-Turbo with ReNO. This highlights the potential of test-time/online optimization compared to traditional fine-tuning, since it can generalize much better to unseen prompt distributions.

Table 7: Comparison of ReNO and Direct Preference Optimization (DPO) with a SDXL-based model. SDXL Base result taken from [13].

| Method | Attribute Binding | | | Aesthetic |
|---|---|---|---|---|
| | Color ↑ | Shape ↑ | Texture ↑ | |
| SDXL Base [69] | 0.6369 | 0.5408 | 0.5637 | 5.604 |
| DPO-SDXL [92] | 0.6793 | 0.5316 | 0.6513 | 5.687 |
| SDXL-Turbo + ReNO | **0.7800** | **0.5955** | **0.7396** | **6.024** |

## B.2  Compositional Text-to-Image Methods.

We show the results for several methods that have been tailored to compositional Text-to-Image generation in Table 8. These methods either explicitly finetune the model for improved compositional generation, or modify the inference process, or repeat the sampling over multiple iterations. We see that ReNO consistently outperforms specific methods tailored for this task.

We also note that some methods use LLMs and other tools (image-editing, customization etc.) to plan out or correct generations [25, 52, 94]. However, these methods significantly impact the generation process through, e.g., iterative generation and planning. In contrast, ReNO only changes the initial noise and doesn't alter the generative model at all. Thus, our method could also be incorporated into these tools to further improve performance.

Table 8: **Quantitative Results on T2I-CompBench**. Full comparison against different Compositional Text-to-Image methods. The best value is bolded, and the second-best value is underlined. Results for compositional methods taken from [66, 94].

| Model | Attribute Binding | | | Object Relationship | | Complex↑ |
|---|---|---|---|---|---|---|
| | Color ↑ | Shape↑ | Texture↑ | Spatial↑ | Non-Spatial↑ | |
| SD2.1 | 0.5065 | 0.4221 | 0.4922 | 0.1342 | 0.3096 | 0.3386 |
| + Composable Diffusion [56] | 0.4063 | 0.3299 | 0.3645 | 0.0800 | 0.2980 | 0.2898 |
| + Attn-Mask-Control [93] | 0.4119 | 0.4649 | 0.4505 | 0.1249 | 0.3046 | 0.3779 |
| + StructureDiffusion [24] | 0.4990 | 0.4218 | 0.4900 | 0.1386 | 0.3111 | 0.3355 |
| + TokenCompose [96] | 0.5055 | 0.4852 | 0.5881 | 0.1815 | 0.3173 | 0.2937 |
| + Attn-Exct [10] | 0.6400 | 0.4517 | 0.5963 | 0.1455 | 0.3109 | 0.3401 |
| + GORS [37] | 0.6603 | 0.4785 | 0.6287 | 0.1815 | **0.3193** | 0.3328 |
| SD-Turbo + ImageSelect [41] | 0.7222 | 0.5552 | 0.6919 | 0.2216 | 0.3154 | 0.4618 |
| (1) PixArt-$\alpha$ DMD | 0.3824 | 0.3414 | 0.4691 | 0.1906 | 0.3060 | 0.3643 |
| **(1) + ReNO (Ours)** | 0.6454 | 0.5658 | 0.7186 | 0.2508 | 0.3138 | 0.4554 |
| (2) SD-Turbo | 0.5513 | 0.4448 | 0.5690 | 0.1739 | 0.3101 | 0.4052 |
| **(2)+ ReNO (Ours)** | 0.7830 | 0.6244 | 0.7466 | 0.2235 | 0.3161 | **0.4829** |
| (3) SDXL-Turbo | 0.6149 | 0.4366 | 0.6001 | 0.2401 | 0.3118 | 0.4250 |
| **(3)) + ReNO (Ours)** | 0.7800 | 0.5955 | 0.7396 | 0.2551 | 0.3147 | 0.4690 |
| (4) HyperSDXL | 0.6535 | 0.4956 | 0.6496 | 0.2509 | 0.3108 | 0.4582 |
| **(4) + ReNO (Ours)** | **0.7904** | **0.6324** | **0.7671** | **0.2616** | 0.3145 | 0.4766 |

**Comparison to Multi-step Noise Optimization Methods.** In Section 4.6, we report quantitative comparison between the multi-step noise optimization method DOODL and ReNO. In Table 9, we additionally report more details on the difference in efficiency between DOODL and ReNO. Note that for the same objective and model family ReNO is 120x faster compared to DOODL.

Table 9: Computational cost comparison of ReNO compared to DOODL.

| Model | sec/iter (total) | T2I-CompBench | VRAM | #params |
|---|---|---|---|---|
| SD2.1 + DOODL (CLIP) | 24s (20min) | 83.33 A100 days | 40GB | 860M |
| SD-Turbo + ReNO (only CLIP) | 0.2s (10s) | 0.63 A100 days | 10GB | 860M |
| SD-Turbo + ReNO | 0.4s (20s) | 1.25 A100 days | 15GB | 860M |

## C   Analysis of Reward Models

We show all results for all combinations of the reward models in Table 10. Broadly, adding all reward models ensures that a meaningful improvement is achieved both on attribute binding and on the aesthetic score. In addition to this, we perform a leave-one-out analysis on Parti-Prompts in Table 11, where one reward is excluded from ReNO and subsequently analyzed.

Even when a particular reward is not optimized for, we see that there is a consistent improvement in the metrics, and in most cases, at least 80% of the images improve even on the left-out reward. This phenomenon across a variety of models (e.g. CLIP, BLIP) trained on differing datasets certainly indicates that there are significant improvements made by ReNO across most of the images. While the reward increase and the percental improvement, can differ based on the one-step model, CLIPScore and ImageReward seem to be less correlated to the other rewards, which could be explained based on the similar backbone employed by HPSv2 and PickScore. Interestingly, PixArt-$\alpha$ DMD achieves the highest reward scores after optimization, which does not follow the quantitative results for T2I-CompBench and GenEval as reported in Section 4.2.

Table 10: Full results for all different reward model combinations considered in ReNO over the attribute binding categories of T2I-CompBench and the LAION aesthetic score predictor [83]. We highlight the best and second-best results per number of reward models.

| Reward Models | Attribute Binding | | | Aesthetic |
|---|---|---|---|---|
| | Color ↑ | Shape ↑ | Texture ↑ | |
| Base (SD-Turbo) | 0.5513 | 0.4448 | 0.5690 | 5.647 |
| + CLIPScore | 0.6625 | 0.5501 | 0.6621 | 5.475 |
| + HPSv2 | 0.6443 | 0.5451 | 0.6859 | 5.752 |
| + ImageReward | 0.7720 | 0.6104 | 0.7334 | 5.611 |
| + PickScore | 0.6341 | 0.5069 | 0.6242 | 5.711 |
| + CLIPScore + HPSv2 | 0.6691 | 0.5664 | 0.6979 | 5.714 |
| + CLIPScore + ImageReward | 0.7749 | 0.6218 | 0.7415 | 5.579 |
| + HPSv2 + ImageReward | 0.7710 | 0.6228 | 0.7518 | 5.692 |
| + PickScore + CLIP | 0.6606 | 0.5500 | 0.6735 | 5.615 |
| + PickScore + HPSv2 | 0.6593 | 0.5571 | 0.6766 | 5.776 |
| + PickScore + ImageReward | 0.7798 | 0.6298 | 0.7354 | 5.662 |
| + CLIPScore + HPSv2 + ImageReward | 0.7735 | 0.6238 | 0.7524 | 5.677 |
| + PickScore + CLIPScore + HPSv2 | 0.6886 | 0.5599 | 0.7012 | 5.733 |
| + PickScore + CLIPScore + ImageReward | 0.7797 | 0.6218 | 0.7513 | 5.620 |
| + PickScore + HPSv2 + ImageReward | 0.7778 | 0.6298 | 0.7457 | 5.713 |
| + All | 0.7830 | 0.6244 | 0.7466 | 5.704 |

**Reward Weighting.** The four reward models that we employ output scores in different ranges. Specifically, HPSv2 mostly ranges between 0.2-0.4, while PickScore is in the range of 20-30 for

most of the images. ImageReward is in the range of $-2$ to $+2$ for the majority of images, and CLIPScore is between 0 and 1. For all our experiments, we use weights of 1.0 for ImageReward, 5.0 for HPSv2, 0.05 for PickScore, and 1.0 for CLIPScore. When each score range is scaled to $[0, 1]$, then these weights correspond to 4.0 for ImageReward, 1.0 for HPSv2, 0.5 for PickScore, and 1.0 for CLIPScore. These weights ensure that the losses from each reward model are roughly similar, with a higher emphasis on ImageReward.

Table 11: Leave-one-out reward evaluation on Parti-Prompts. The listed reward in the first column is left out in ReNO and subsequently analyzed with respect to its change as well as the percentage of generations where ReNO improves this reward.

| | Initial Reward ↑ | ReNO Reward↑ | Change ↑ | Improve % ↑ |
|---|---|---|---|---|
| **PixArt-$\alpha$ DMD** | | | | |
| CLIPScore [0, 1] | 0.332 | 0.386 | +0.054 | 95.1 |
| PickScore [20, 30] | 22.235 | 23.788 | +1.553 | 97.6 |
| HPSv2 [0.2, 0.4] | 0.281 | 0.324 | +0.043 | 97.7 |
| ImageReward [-2, 2] | 0.896 | 1.367 | +0.471 | 88.7 |
| **SD-Turbo** | | | | |
| CLIPScore [0, 1] | 0.353 | 0.386 | +0.033 | 80.4 |
| PickScore [20, 30] | 22.028 | 23.232 | +1.204 | 91.1 |
| HPSv2 [0.2, 0.4] | 0.266 | 0.310 | +0.044 | 94.7 |
| ImageReward [-2, 2] | 0.552 | 1.243 | +0.691 | 90.3 |
| **SDXL-Turbo** | | | | |
| CLIPScore [0, 1] | 0.360 | 0.379 | +0.019 | 74.2 |
| PickScore [20, 30] | 22.505 | 23.185 | +0.680 | 82.3 |
| HPSv2 [0.2, 0.4] | 0.280 | 0.310 | +0.030 | 93.4 |
| ImageReward [-2, 2] | 0.920 | 1.270 | +0.350 | 81.0 |

# D   User Study

We perform our user study on Amazon Mechanical Turk, and pay participants based on prior guidelines [65], which also ensures the compensation is above the minimum wage. We use pairwise preferences due to its simplicity, allowing users to mark ties between images that are equally good/bad. Each pairwise comparison is treated as an individual entity and handed to an individual user to minimize user biases. In particular, each pairwise comparison between the two models has involved at least 339 unique users (and 673 maximal), with the average being 495. To reduce the number of user comparisons, we perform the user study on a subset of Parti-Prompts totaling slightly above 1000 prompts (excluding challenges ['*Basic', 'Imagination', 'Perspective', 'Linguistic Structures'*] and categories ['*Abstract', 'Indoor Scenes', 'Produce & Plants'*]).

We ask users to answer the following three questions:

- **On personal preference:** "Which image would you **personally prefer** getting given the input text (based on your personal tradeoff between faithfulness and aesthetics)?"
- **On aestheticness:** "Which image do you find more aesthetically pleasing?
- **On faithfulness:** "Which image is more faithful to the input text?"

We also provide additional information on terminology:

- **Faithfulness:** The generated image should reflect all key concepts, their relations and their attributes given in the text prompt.
- **Aestheticness:** Refers to the style, coloring and interpretation in the depiction of concepts (i.e. "looks better").
- **Personal preference:** Some generations can be more faithful, but less aesthetic, or the other way around. Choose which you prefer :).

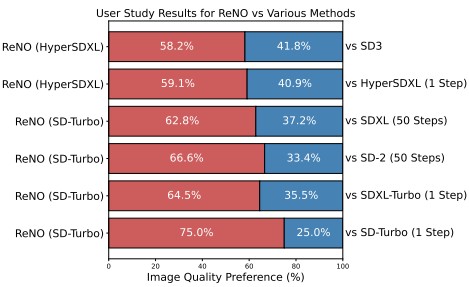

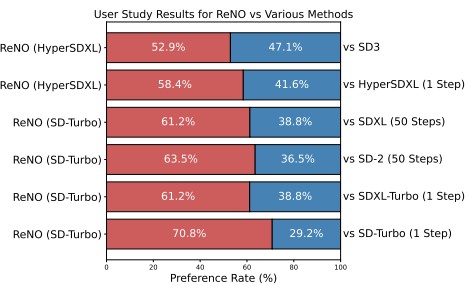

(a) User Preferences for Image Quality  (b) User Preferences for Prompt Faithfulness

Figure 11: User study results on aestheticness and faithfulness based on Parti-Prompts.

For the competing methods, we use default parameters: one-step generation without classifier-free guidance (CFG), and CFG = 7.5 for SD2.1 and CFG = 5.0 for SDXL. For the proprietary SD3, we generate images through the API provided at `https://platform.stability.ai/`. Note that to compare SDXL with SD-Turbo + ReNO, we generate images in $1024 \times 1024$ for SDXL as this is its native resolution and then afterward downsize them to $512 \times 512$.

In Figure 11, we report the results for the specific questions on faithfulness and aestheticness. Interestingly, for all models, the preference for aestheticness is even larger than that of faithfulness. While the quantitative results on T2I-CompBench and GenEval reported in Section 4.2 mainly benchmark the prompt following improvements of ReNO, this result confirms ReNO's benefits in improving the general quality of generated images.

## E  Implementation Details

Our code is built with `Pytorch` [67] and is mainly based on the `diffusers` library [90]. It is available at `https://github.com/ExplainableML/ReNO`.

### E.1  Algorithm

We outline the overall algorithm for ReNO in Algorithm 1. Note that, we choose gradient ascent with Nesterov momentum as we found this for our computational budget to yield the best results. Although line-search-based methods such as L-BFGS are viable options [5], we find that even without them gradient ascent provides efficient and effective optimization of the criterion function. However, L-BFGS or gradient ascent without momentum might also be viable optimization methods for ReNO.

---

**Algorithm 1** ReNO

---

**Input:** p (prompt), $G_\theta$ (One-Step T2I Model), $\mathcal{R}_\psi^{0,1\ldots n}$ (Reward Functions), $\lambda_{0,1\ldots n}$ (Reward Weights), $m$ (# Optimization Steps), $\eta$ (Learning Rate), $\lambda_{\mathrm{reg}}$ (Regularization Strength)

Initialize $v_{-1} = 0.0$, $\varepsilon^0 = \mathcal{N}(0, \mathbf{I})$, $R^\star = -\inf$.

**for** $t = 0$ **to** $m$ **do**

    Generate image $\mathbf{x}_0^t = G_\theta(\varepsilon^t, p)$

    Compute reward-based criterion $R^t = \sum_i^n \lambda_i \mathcal{R}_\psi^i(\mathbf{x}_0^t, \mathrm{p})$

    $\mathrm{grad}_t = \nabla_{\varepsilon^t}[\lambda_{\mathrm{reg}} K(\varepsilon^t) + R^t]$

    $\mathrm{grad}_t = \mathrm{GradNormClip}(\mathrm{grad}_t, 0.1)$

    $v_t = 0.9 \cdot v_{t-1} + \eta \cdot \mathrm{grad}_t$

    $\varepsilon^{t+1} = \varepsilon^t + v_t$

    **if** $R^t > R^\star$ **then**

        $\mathbf{x}_0^\star = \mathbf{x}_0^t$, $R^\star = R^t$

    **end if**

**end for**

**return** $\mathbf{x}_0^\star$

---

### E.2 ReNO hyperparameters

As detailed in Algorithm 1 the main hyperparameters in ReNO are the learning rate $\mu$, the regularization strength $\lambda_{\text{reg}}$ and the choice for reward models, which we explore in Appendix C. We use $\lambda_{\text{reg}} = 0.01$ for all our experiments. For the learning rate, we use $\mu = 5$ for all our $512 \times 512$ models and $\mu = 10$ for HyperSDXL that generates $1024 \times 1024$ as we found this to give a good balance between exploration, improvements, and fast convergence. Note that in combination with gradient norm clipping, this also prevents major changes in the noise that would completely change the generated image. This effect can be observed in Figures 1 and 6, and Appendix A, as the image after ReNO optimization still shares significant details with the initially generated image.

### E.3 Models

SD-Turbo, SDXL-Turbo [81], and HyperSDXL [75] are built with a UNet [77] architecture similar to the one proposed in Rombach et al. [76]. On the other hand, PixArt-$\alpha$ DMD [12, 13] leverages a Diffusion Transformer [21, 59, 68] based architecture. We use the checkpoints of SD-Turbo, SDXL-Turbo, HyperSDXL, and PixArt-$\alpha$ DMD supplied through `huggingface`. For HyperSDXL, we use the one-step UNet checkpoint (as opposed to the LoRA version).

### E.4 Rewards

In this work, we employ the four following reward models for ReNO.

#### E.4.1 Human Preference Score v2 (HPSv2)

HPSv2 [97] is an improved version of the HPS [98] model, which uses an OpenCLIP ViT-H/14 model and is trained on prompts collected from DiffusionDB [95] and other sources. Note that here we employ the further improved HPSv2.1 checkpoint.

#### E.4.2 PickScore

PickScore also uses the same ViT-H/14 model, however is trained on the Pick-a-Pic dataset which consists of 500k+ preferences that are collected through crowd-sourced prompts and comparisons.

#### E.4.3 ImageReward

ImageReward [100] trains a MLP over the features extracted from a BLIP model [49]. This is trained on a dataset of images collected from the DiffusionDB [95] prompts.

#### E.4.4 CLIPScore

Lastly, we use CLIPScore [35, 73], which was not designed specifically as a human preference reward model. However, it measures the text-image alignment with a score between 0 and 1. Thus, it offers a way of evaluating the prompt faithfulness of the generated image that can be optimized. We use the model provided by OpenCLIP [38] with a ViT-H/14 backbone.

### E.5 Metrics

Apart from the user study (details in Appendix D) and the reward models themselves in Table 11, we benchmark ReNO with three different evaluation schemes as detailed in the following.

#### E.5.1 T2I-CompBench

T2I-CompBench is a comprehensive benchmark proposed by Park et al. [66] for evaluating the compositional capabilities of text-to-image generation models. The benchmark consists of three categories and six sub-categories of compositional text prompts: (1) Attribute binding, which includes color, shape, and texture sub-categories, where the model should bind the attributes with the correct objects to generate the complex scene; (2) Object relationships, which includes spatial and non-spatial relationship sub-categories, where the prompts contain at least two objects with specified relationships; and (3) Complex compositions, where the prompts contain more than two objects

or more than two sub-categories. The attribute binding subtasks are evaluated using BLIP-VQA (i.e., generating questions based on the prompt and applying VQA on the generated image), spatial relationships are evaluated using an object detector, non-spatial relationships are evaluated through CLIPScore (CLIP ViT-B/32), and complex compositions are evaluated using all three models.

### E.5.2 GenEval

GenEval is an object-focused framework introduced by Ghosh et al. [28] for evaluating the alignment between text prompts and generated images from Text-to-Image (T2I) models. Unlike holistic metrics such as FID or CLIPScore, GenEval leverages existing object detection methods to perform a fine-grained, instance-level analysis of compositional capabilities. The framework assesses various aspects of image generation, including object co-occurrence, position, count, and color. By linking the object detection pipeline with other discriminative vision models, GenEval can further verify properties like object color. All the metrics on the GenEval benchmarks are evaluated using a MaskFormer object detection model with a Swin Transformer [58] backbone. Lastly, GenEval is evaluated over four seeds and reports the mean for each metric, which we follow.

### E.5.3 LAION Aesthetic Score Predictor

Furthermore, we employ the improved LAION Aesthetic Predictor as an evaluation metric. It consists of an MLP trained on top of a CLIP [73] backbone. Importantly, this predictor does not take the prompt as a joint input with the image. Thus, the aestheticness of an image is always evaluated independently of what prompt was used to generate it. This predictor can also be used as an objective to improve the aesthetic quality of generated images, which we briefly investigated. We found that while numerically, the generated images achieve a higher score, their actual visual quality does not seem to always be higher. We hypothesize that this is because the predictor is independent of the given prompt and thus might be more prone to reward-hacking.

### E.6 Diversity Analysis

We generated images with 50 different seeds for 10 prompts from each of the 11 challenges of PartiPrompts, totaling 110 prompts. Then, for each prompt, we evaluate the diversity over the 50 seeds by computing the mean pairwise LPIPS [106] and DINO [9, 64] scores. The higher these two scores are, the less diverse the generated images across seeds. We report the mean and standard deviation across all prompts.

### E.7 Comparison to DOODL

For our comparison to multi-step noise optimization DOODL, we use the first 50 prompts from each of the attribute binding categories of T2I-CompBench. We benchmark DOODL using the official codebase (https://github.com/salesforce/DOODL/blob/main/doodl.py), adapted to SD2.1 with 50 steps. We chose to focus on the SD2.1 model family because when running DOODL on SDXL, it exceeds 40GB of VRAM making it unfeasible for single GPU runs and thus inference.

### E.8 FLUX-schnell results

We find that noises employed for FLUX-schnell with one step translate very well to FLUX-schnell wtih four steps. Thus, due to efficiency we apply ReNO to FLUX-schnell with one step and afterward feed in the optimal noise to the four step FLUX-schnell model to obtain our final generation. Due to VRAM constraints, we generate samples in $512 \times 512$ including CPU-offloading such that FLUX-schnell + ReNO runs within 40GB of VRAM. The FLUX-dev results reported in Table 3 are taken from Liu et al. [55]. We report FLUX-schnell + ReNO results on the attribute binding categories of T2I-CompBench in Table 12 and a qualitative comparison in Figure 12.

## F Broader Impact

Text-to-Image models have a wide variety of uses in different settings. While they can be used for harmful purposes, practcal deployments of these models (including ours) must be made with a safety checker/filter to prevent the generation of NSFW content. In our work, we rely on existing pretrained

Table 12: Comparison of $512 \times 512$ FLUX-schnell with and without ReNO on the attribute binding categories of T2I-CompBench.

| | Attribute Binding | | |
|---|---|---|---|
| | Color ↑ | Shape ↑ | Texture ↑ |
| FLUX-schnell | 0.69 | 0.53 | 0.67 |
| **FLUX-schnell + ReNO** | **0.80** | **0.60** | **0.75** |

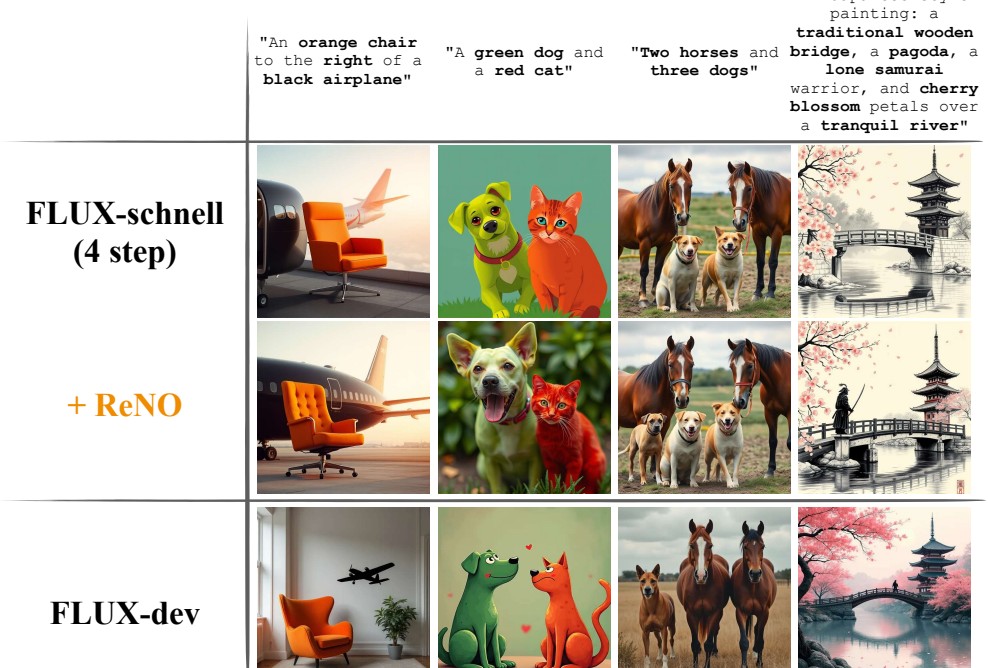

Figure 12: Non-cherry-picked results ($seed = 0$) for FLUX-schnell with and without ReNO compared to FLUX-dev.

models, and therefore would inhereit its biases. However, we believe that our reward optimization framework is flexible to also include safety and fairness and potential objectives which would be an option to mitigate the harms of existing image generation models.

