# OpenReview forum: "ReNO: Enhancing One-step Text-to-Image Models through Reward-based Noise Optimization"
_NeurIPS.cc/2024/Conference — NeurIPS 2024 poster_

### Official Review · Reviewer_EAgt · 2024-07-07

**Soundness:** 2
**Presentation:** 2
**Contribution:** 1
**Rating:** 3
**Confidence:** 4

**Summary:**

In this work, the authors propose finetuning the noise that a one-step diffusion model predicts a clean image from, with respect to an ensemble of preference constraints.  Because only a one-step diffusion model is used, optimizing the noise is fast in terms of number of steps (and therefore wall clock time).  The authors demonstrate that the performance of this optimized one-step text-to-image model is comparable to diffusion models that utilize multiple levels of denoising on preference benchmarks.

**Strengths:**

The strength of this paper is that the approach is intuitive, and the success of the approach is reasonable.

**Originality:** The paper does not appear to be very novel.  Optimizing the initial source noise has been explored before, and the authors are reusing existing preference classifiers/constraints.  The only ``novel" component seems to be applying it to a one-step diffusion model, which may not actually be a strength (see weaknesses), which seems more as a simplified edge case, as well as utilizing multiple constraints in a weighted fashion which has limited originality.

**Quality:** The paper quality is not particularly high.  The organization of the paper is rather messy.  The evaluations are also insufficient to demonstrate the benefits of their proposed approach.

**Clarity:**  The clarity could be improved.  For example, Equation 7 was referenced before it was even written.  Furthermore, the background sections were structured strangely.  The section of "Background: One-Step Diffusion Models" spends most of its passage discussing regular Diffusion Models instead; one-step diffusion models were only mentioned in two sentences (lines 116-117).
 Furthermore, Section 2 - titled "ReNO", reads almost as a background/related works section (most clearly demonstrated in lines 118-125), but ReNO is only introduced two pages later (approx line 175).  Then, there is a separate Related Works section.  The paper could benefit from severe reorganization to improve its clarity.

**Significance:** In this reviewer's perspective, this paper has limited significance.  The results seem expected, and do not provide deep insight (they essentially verify that optimization of the the initial noise helps performance), and the approach is only limited to one-step diffusion models and not the diffusion modeling in general.  Furthermore, the reliance on an ensemble means that this approach does not work on a singular specific preference of interest, but only on the average of multiple.

**Weaknesses:**

First and foremost, the comparisons showcased within this paper are flawed.  In Table 2 and Table 3, the authors compare their approach against the performance of default text-to-image models.  It should be almost expected that the performance of ReNO, by virtue of task-specific optimization, **should** have improved performance over default text-to-image models.  This is therefore not an interesting comparison; in fact, it even raises suspicions on the approach in the cases that it does not outperform default models (e.g. Attribute Binding in Dall-E 3).  The authors should instead compare their optimized approach against other optimized approach; for example, DOODL, or DOODL modified to be one-step, the approach from Samuel et al., etc.  This provides a clearer picture of ReNO as an optimization scheme compared to other optimization schemes in tackling preference-respecting generation.  The comparison to DPO is a nice small start, but we need more such comparisons - across benchmarks, and across more task/metrics.  It is really strange that the authors do not compare to the works they state are the most related to their approach.

The other weakness of this approach is that it is limited to one-step diffusion models.  As the authors state, "backpropagating the gradient through multiple denoising steps can lead to exploding/vanishing gradients, rendering the optimization process unstable."  They instead limit themselves to using only "a distilled one-step T2I model".  The authors have not solved a fundamental problem in a general way, of how to optimize the noise for general text-to-image models; instead, they only demonstrate it for a single-step T2I model.  The impact of this work is therefore extremely limited; such insights can clearly not generalize to arbitrary diffusion models; and the authors **have not solved** a key issue of the approach of noise optimization but just ignored it entirely in favor of a more limited problem setting where the results they achieved is rather expected.

The method also seems to rely heavily on an ensemble.  However, this makes it rather ungeneralizable; under this approach it is not possible to optimize the text-to-image model with respect to one specific, particular preference.  Instead, the reliance on multiple reward functions simultaneously implicitly means that the resulting policy will balance between each of the functions used in the ensemble.  If there were a novel preference that it were important to optimize a text-to-image model to respect, this approach would not be able to be applied successfully.

This reviewer found that this work lacks severely in insight - indeed, many of the purported results are completely to be expected.  The authors repeatedly tout the fast training benefits of ReNO, but really this boils down to it being forced to work with a one-step diffusion model, which makes the fast optimization a rather expected result.  The computational cost was not improved in any way by the authors by their approach, it comes for free due to them restricting their **choice** of diffusion model to a one-step one.  Therefore, it is completely unsurprising that using a one-step diffusion model would result in faster optimization than multi-step ones; there is no new insight to be gained here.  Furthermore, the user study was conducted for SD-Turbo + ReNO against default models like SD-Turbo without any optimization; of course, there should be an improvement!  All these results have verified is that optimizing the noise helps, but this is to be expected - it is completely unsurprising.

Furthermore, a severe weakness of ReNO is that once optimized, there is no diversity in the output - particularly because the approach is one-step.  For multi-step diffusion, even with the initial noise being optimized, the resulting output has variability because the other denoising steps have stochasticity (of resampling the Gaussian noise).  However, one-step diffusion has no diversity for a fixed (optimized) noise.  Even though it may be cheaper in wall-clock time to optimize one one-step image, if a batch of $n$ images were to be generated ReNO would need $n$ times the proposed time cost for one image.  On the other hand, Samuel et. al only optimize the initial generation point for multi-step diffusion, enabling a result that still has diversity while still respecting preference.  The speed is by default 1-4 minutes for Samuel et . al which is comparable or even preferable to ReNO, particularly because it can generate a batch all at once whereas ReNO needs to be reoptimized for each image which may take longer.  Also, Samuel et. al reports optimization improvements that reduce it from minutes to seconds, which makes it strictly better than RENO which takes seconds to just generate one image that has no diversity. Samuel et. al state in Section 6.1 that it takes 1-5 minutes to adapt to the new concept and 1-2 seconds to generate new semantically correct images.

The authors do not supply new criteria, nor do they innovate the approach of optimizing the noise.  Instead, they simply apply multiple existing criteria simultaneously.  Furthermore, the approach is limited to one-step diffusion models, which inhibits its generality.  Ultimately, the scope of this work feels more like a workshop paper rather than a conference-level paper.

**Questions:**

Are there ways to optimize multiple noises through ReNO to generate a batch of preference-respecting images without essentially optimizing each noise separately?

What are the ways ReNO can enable or address the backpropagation through multiple denoising levels, and provide interesting implications for general multi-step diffusion models?  Are there ways this can be demonstrated in the rebuttal?

**Limitations:**

This reviewer foresees no substantial potential negative societal impact from this work.  However, in the Limitations section, the authors state that they hypothesize that the reward models may be limiting, and that stronger reward models and preference data may be crucial in enhancing results further.  This reviewer feels like this hypothesis could be directly tested within the scope of this work; building off of the initial results of Table 1, further study could be performed on utilizing subsets of the complete set of reward functions to evaluate the benefits of each.

---

> ### Author Rebuttal · Authors · 2024-08-07
>
> We would like to clarify the contributions of ReNO to ensure the problems ReNO is aiming to solve are understood. We tackle the question of whether we can **generally** enhance T2I models *without any fine-tuning* at test time. We propose to tackle this through the Noise Optimization framework by considering *human-preference reward models* as the optimization criterion. This is not a task-specific framework, nor are we proposing a new general method for noise optimization. Instead, we are specifically aiming to generally enhance a trained model at test time through our approach. This has not yet been considered or discussed in previous work, and how this generally performs was unclear. Additionally, it poses challenges that we propose to tackle through the use of one-step models (for computational efficiency), the use of multiple reward models (to prevent reward hacking), and by using noise regularization (for the noise to stay in distribution).
>
> We thoroughly evaluate ReNO across four one-step T2I models (i.e SD-Turbo, SDXL-Turbo, Pixart-$\alpha$-DMD, and HyperSDXL), across 3 benchmarks (T2I-Compbench, GenEval, Parti-Prompts), human preference evaluations (nearly 10k comparisons), and now also the diversity of the outputs. Our results show generally enhanced performance **without** optimizing for any specific task. For example, SD-Turbo is elevated to prompt-following levels close to DALLE-3 and is significantly preferred over 50-step SDXL, while HyperSDXL + ReNO is preferred over SD3 (8B). This substantial performance increase is a major contribution of ReNO, and we argue it is a **very significant** finding. It elucidates the importance of initially sampled noise and demonstrates its effective manipulation within a reasonable timeframe. We believe it is fascinating and unexpected that such a lightweight approach can significantly enhance the performance of image generation. This also differentiates ReNO from previous noise optimization work by showcasing the framework's power in a new, more general setting. Lastly, it motivates further research into understanding and controlling initial noise and provides a novel way of benchmarking future T2I reward models.
>
> > ***Comparison to SeedSelect (Samuel et al.)***
>
> SeedSelect tackles rare-concept generation, a fundamentally different problem from the one ReNO is solving. I.e. the goal of SeedSelect is, given 3-5 reference images, to generate images of rare concepts represented in these images. They propose to tackle this based on the noise optimization framework and reduce computational time, as mentioned in our related work section: "To mitigate this, Samuel et al. [64] propose a bootstrap-based method to increase the efficiency of generating a batch of images. However, this method is limited to settings where the goal is to generate samples including a concept jointly represented by a set of input images". On the other hand, we consider a more general setting, where each image is generated separately, given noise and text as input. Therefore, adapting SeedSelect to the general setting considered in our paper is not straightforward.
>
> > ***The authors demonstrate that the performance of this optimized one-step text-to-image model is comparable to diffusion models that utilize multiple levels of denoising on preference benchmarks.***
>
> We disagree with this statement. We clearly show across multiple benchmarks and user studies that ReNO-optimized one-step models significantly outperform their corresponding multi-step models in all benchmarks and even their next-generation multi-step ones (See Tables 2, 3 & 5 and Figures 4 & 5).
>
> > ***Optimizing the initial source noise has been explored before, and the authors are reusing existing preference classifiers/constraint.***
>
> As mentioned above, while we are repurposing existing reward models, as far as we are aware, the optimization criterion of human-preference reward models for noise optimization has not been considered or discussed before.
>
> > ***The results seem expected, and do not provide deep insight (they essentially verify that optimization of the the initial noise helps performance)***
>
> We would argue that the magnitude of the performance increase makes it a **very significant** insight that is not expected at all. One-step models are known to have significantly worse visual quality and prompt following than the base multi-step models, therefore it is not expected that purely optimizing the initial noise would make them outperform even next generation multi-step models.  Further, increasing the diversity of the generated images compared to one-step models is also unexpected and a valuable finding as less diversity is also a common weakness of one-step models.
>
> > ***First and foremost, the comparisons showcased within this paper are flawed. In Table 2 and Table 3, the authors compare their approach against the performance of default text-to-image models. It should be almost expected that the performance of ReNO, by virtue of task-specific optimization, **should** have improved performance over default text-to-image models.***
>
> As mentioned, we are not doing any task-specific optimization. We are proposing one optimization that **generally** improves a T2I model, and we thoroughly evaluate each ReNO-enhanced model over different benchmarks as well as human evaluation.
>
> > ***This is therefore not an interesting comparison; in fact, it even raises suspicions on the approach in the cases that it does not outperform default models (e.g. Attribute Binding in Dall-E 3).***
>
> We respectfully disagree. Pushing SD-Turbo to a performance close to DALL-E 3 just by changing the initial noise is not expected and is actually a very interesting comparison as also acknowledged by Reviewer UuFQ.

---

> ### Author Response · Authors · 2024-08-07
> **Rebuttal by Authors [2/2]**
>
> > ***the authors **have not solved** a key issue of the approach of noise optimization but just ignored it entirely in favor of a more limited problem setting where the results they achieved is rather expected.***
>
> While we agree that solving the issue of computational efficiency for multi-step generation with noise optimization is a very interesting research question, we never claim to solve this problem generally with ReNO. We sidestep this challenge by considering one-step diffusion models. Note that this is still a major insight because applying noise optimization in the form of DOODL/D-Flow is not practical for general T2I generation, as mentioned in the global rebuttal.
>
> > ***Furthermore, the reliance on an ensemble [...] ungeneralizable [...]***
>
> We report the performance of all the combinations of reward models in Table 1 & 7. As can be seen, also without an ensemble with only HPSv2 (better image quality) or ImageReward (better prompt following), ReNO achieves significant performance improvements. We are unsure if we understand why this would make ReNO not generalizable. In the general setting we are considering, the goal is no one specific preference but a general improvement of the used model.
>
> > ***If there were a novel preference that it were important to optimize a text-to-image model to respect, this approach would not be able to be applied successfully.***
>
> On the contrary, our approach is designed to be flexible and adaptable to various objectives. There's no inherent limitation preventing the application of ReNO to novel preferences or optimization goals. In fact, we demonstrate in Figure 3 how "personalized" objectives can be effectively incorporated. With just 10 optimization iterations, we show significant increases in specific color attributes (redness/blueness) of generated images.
>
> > ***Furthermore, the user study was conducted for SD-Turbo + ReNO against default models like SD-Turbo without any optimization; of course, there should be an improvement!***
>
> In the user study, we specifically compare SD-Turbo + ReNO against SDXL-Turbo, SD2.1 (50-step), and SDXL (50-step). All of these models are usually preferred over SD-Turbo. SD-Turbo + ReNO is significantly preferred over all of them, even the next-generation SDXL with 50 steps! Additionally, HyperSDXL + ReNO is preferred over SD3 (8B).
>
> > ***Are there ways to optimize multiple noises through ReNO to generate a batch of preference-respecting images without essentially optimizing each noise separately?***
>
> Theoretically, this could also be incorporated into ReNO as a further Criterion function. However, the problem we are tackling is general T2I generation, which is by default only per single prompt and noise. Thus, this is out of the scope of our work but an interesting future direction.
>
> > ***What are the ways ReNO can enable or address the backpropagation through multiple denoising levels, and provide interesting implications for general multi-step diffusion models? Are there ways this can be demonstrated in the rebuttal?***
>
> While we agree that solving the challenges of noise optimization for multi-step generation is a very interesting research question, we never claim to solve this problem generally with ReNO, and it is also out of the scope of this work. We show how the noise optimization network can be leveraged to effectively arrive at a **very significantly** better model. We leave how to best adapt our findings to multi-step diffusion models to future work.
>
> > ***Equation 7 was referenced before it was even written***
>
> Thanks for this pointer, it was supposed to refer to Equation 4 and we will update it in the paper accordingly.
>
> > ***one-step diffusion models were only mentioned in two sentences (lines 116-117).***
>
> The paragraph [lines 112-125] completely serves to introduce the different one-step diffusion models we employ in this work and how they work in general.
>
> > ***reads almost as a background/related works section (most clearly demonstrated in lines 118-125)***
>
> In this part, we introduce the four different one-step models we use to benchmark ReNO. We consider this as background for ReNO, as the section is also titled.
>
> > ***Then, there is a separate Related Works section. The paper could benefit from severe reorganization to improve its clarity.***
>
> Thank you for this comment; we will consider how to improve the clarity. With the separate related work section after the introduction of ReNO we aim to contextualize ReNO within the scope of all related work. Which part of the way we introduce ReNO exactly was unclear? We would be happy to incorporate any specific suggestions to enhance the clarity.

---

> > ### Comment · Reviewer_EAgt · 2024-08-14
> > **Reviewer Response [1]**
> >
> > This reviewer appreciates the detailed rebuttal, and provides thoughts in response:
> >
> > This reviewer understands that exploring the optimization of the noise vector is the main problem ReNO is aiming to solve.  A limitation is that, the analysis seems only to hold for one-step models.  It is not obvious that optimizing the noise is generally useful for diffusion modeling; especially since over multiple timesteps, the effect of each particular noise sample is minimized - potentially including the initial one.  The impact of this work is therefore severely limited.
> >
> > To that note, it is not particularly impressive if by default, one-step T2I models are capped in terms of modeling capability.  If even with the best optimization, T2I models cannot outperform multistep diffusion models without optimization (e.g. DALL-E 3), then one-step T2I models are essentially a dead-end, barring amazing new developments in the space.  This reviewer therefore disagrees that this is a particularly significant finding.  The purported importance of initially sampled noise appears only to hold for this limited, toy example and is not supported for diffusion models in general.  The authors do not provide real insights into how to perform noise optimization over multiple timesteps, nor can they provide real insights into noise optimization for the general case of (potentially multi-step) diffusion models as the usefulness of the source noise is then in question.
> >
> > In Tables 2 it is pretty apparent that DALL-E 3 outperforms ReNO, while being a multi-step denoising diffusion model.  There is some confusion on why the authors disagree with the initial statement, and suggests that “ReNO-optimized one-step models significantly outperform their corresponding multi-step models in all benchmarks”.  Furthermore, Table 5 doesn’t seem relevant to the discussion.
> >
> > This reviewer would like to clarify that when the authors state that they propose an optimization that “generally improves a T2I model”, they are really referring to an optimization that improves a one-step T2I model only.  It is not obvious that such benefits extend to the multi-step case.

---

> > > ### Comment · Reviewer_EAgt · 2024-08-14
> > > **Reviewer Response [2]**
> > >
> > > A major point of contention is that the authors have “side-stepped” a crucial challenge of noise optimization for general T2I models, rather than address it head-on, by limiting their scope to one-step T2I models.  But then, the implications of their findings are severely limited to just one-step cases.  Nothing seems to be able to be generalized to the multi-step case, and no insights can be made about noise optimization for general-purpose T2I models.  But since the capabilities of one-step T2I models are inherently limited, and even with this optimization scheme it cannot outperform default multi-step diffusion models such as DALL-E 3, this reviewer fails to see the discoveries of this work as impactful or useful - especially since the findings in this work cannot generalize to arbitrary T2I model settings.  “Side-stepping” is not something to be celebrated, if it severely limits the implications and impact of their work.  The insight into the behavior of general diffusion modeling is limited, if existent at all.
> > >
> > > Regarding multiple reward models: it still appears that ReNO depends on an ensemble of reward models.  And that performance does not increase significantly without the ensemble.  Relying on an ensemble suggests that this method may not perform well for a specific, novel criteria (in situations where other existing criteria in the ensemble are not as important)
> > > The comment remains the same, that something that has explicit optimization with respect to preference intuitively should outperform something that has not been optimized whatsoever.  This reviewer still believes the user study is unfair, and that it is completely intuitive and expected that an SD-Turbo + ReNO should outperform non-optimized SD-Turbo with respect to preference, because SD-Turbo+ReNO was explicitly optimized to resepct it!  In fact, it is troubling that even with optimization it does not outperform DALL-E 3; intuitively these optimized techniques should always outperform non-optimized ones.
> > >
> > > Generating a batch: once again, to generate a batch of $n$ images ReNO needs to optimize $n$ independent noises; because of the one-step nature of the underlying diffusion model.  Compare this with a multi-step diffusion model, even where the number of levels is 2 - diversity is automatically provided through the resampling of the source noise which adds stochasticity.  Utilizing only a one-step diffusion model prohibits generalization of a batch in a tractable manner and is deeply unsatisfying.
> > >
> > > Optimizing multiple noise levels: by describing the implications of noise optimization for multi-step generation as out of scope, the authors have inherently painted the scope of their work to a very small, unsurprising, and rather insightful scope.  Again, their findings cannot generalize to the common, more powerful multi-step diffusion model case.  Nor is it surprising that an optimized approach beats unoptimized approaches in terms of preference. Furthermore, focusing only on one-step diffusion models introduces a litany of new issues, such as bad default generation behavior and also inability to generate a batch from the same optimized source noise.
> > >
> > > It is strange that the authors are adamant on staying with a one-step diffusion model, refusing to even try some distilled technique that can generate samples in 2 steps or 6 steps (e.g. some LCM models).  This would naturally address complaints about generating batches, would provide initial insights into noise optimization for the general case over multiple timesteps, and the few steps would also hopefully avoid gradient explosion/vanishing.
> > >
> > > The comments regarding writing remain, e.g. one-step diffusion models are only mentioned in lines 116-117, but the entire section of 2.1 is titled One-Step Diffusion models, despite mostly covering general Diffusion Modeling.
> > >
> > > As the majority of the fundamental key concerns this reviewer has with the work remain, this reviewer cannot recommend the paper for acceptance.

---

> ### Author Response · Authors · 2024-08-14
>
> We thank the reviewer for engaging with our rebuttal. We would like to first clarify our motivation for approaching this work: Our goal was to obtain the best possible/extremely high-quality text-to-image generation using open-source models and tools. To this end, models such as the 50-step SD2.1/SDXL models would be the first choice. However, the shortcomings of these models are well-documented and are further corroborated in our paper. Alternately, one could train bigger models on larger datasets with higher-quality data (as is the trend with SD2.1 -> SDXL -> SD3). Unfortunately, these require large-scale GPU resources unavailable to most research groups (e.g. DALL-E 2 itself uses 40000+ A100 GPU days) and additionally, current SOTA models are paid services and not open source (DALLE-3, SD3 (8B)). Therefore, test-time optimization methods are a compelling alternative to enhance the generation quality of existing multi-step T2I models. These methods (e.g. DOODL) work with multi-step models and enhance the quality of the generated images. However, not only are they computationally expensive, they also provide limited improvements in the metrics that they optimize for(on average CLIPScore increases by 0.03 with DOODL on 50 step SD2.1). Our hypothesis was that the lack of effective optimization was due to exploding gradients and other challenges of optimizing multi-step diffusion models. Therefore, we made the unconventional decision of optimizing the initial noise of a one-step model. Apriori, it was unclear if these models could even match the corresponding multi-step model even after noise optimization, let alone surpass them. To our surprise, not only was it 60x faster to optimize, but the gains in CLIPScore were 4x (0.12) that were obtained from optimizing multi-step models (e.g., DOODL). Enhancing this further, we incorporated other models that could provide complementary signals to improve both visual quality (e.g., HPSv2) and prompt following (e.g., ImageReward). As a result, we obtained image generation results that were the highest reported results among any open-source method on T2I-Compbench and GenEval. Further, even for the same time that a 50-step SDXL takes for generation, we show better prompt following by performing noise optimization with SD-Turbo (Fig 5). We believe that providing a recipe to enhance the quality of text-to-image generation (noise optimization of the best-distilled models) in a cost-effective manner is our key contribution.
>
>
> > ***In Tables 2 it is pretty apparent that DALL-E 3 outperforms ReNO, while being a multi-step denoising diffusion model. There is some confusion on why the authors disagree with the initial statement, and suggests that “ReNO-optimized one-step models significantly outperform their corresponding multi-step models in all benchmarks”. [...] In fact, it is troubling that even with optimization it does not outperform DALL-E 3;***
>
> Our finding is for one-step models and their **corresponding** multi-step models. SD-Turbo is based on SD2.1 with an 800M U-Net and a 336M params CLIP ViT-H text encoder, which makes it the multi-step model to compare to. SD-Turbo and DALLE-3 are models trained with very different resources and architectures. While specific details for DALLE-3 are not available, e.g. SD3 leverages a DiT of 8B params with a T5-XXL (4B params) text encoder. We show that in all experiments conducted, SD-Turbo outperforms 50-step SD2.1 and even 50-step SDXL, which is the next-generation multi-step model. Based on our findings, a one-step model based on SD3/DALLE-3, like SD3-Turbo, enhanced with ReNO should outperform multi-step SD3/DALLE-3. Unfortunately, these models are proprietary, and thus, we could not benchmark with SD3-Turbo. To summarize, DALLE-3 and SD3 are "two generations" after SD-Turbo and thus, do not constitute a fair comparison between one-step and multi-step models. Similarly, a method improving LLaMa2-7B isn't expected to outperform GPT-4 to be a meaningful research contribution.
>
> > ***Regarding multiple reward models: it still appears that ReNO depends on an ensemble of reward models. And that performance does not increase significantly without the ensemble. Relying on an ensemble suggests that this method may not perform well for a specific, novel criteria (in situations where other existing criteria in the ensemble are not as important)***
>
> As mentioned in our previous answer, ReNO can be flexibly used based on given preferences. While we benchmark all current T2I reward models we are aware of, adapting this to new models is straightforward as long as the novel criteria are expressed in a differentiable function. See, for example, the color example in Figure 3, or given a new, more robust, and stronger T2I reward model, ReNO can be employed with just this reward. Table 1 does show that already a single reward model achieves significant improvements.

---

> > ### Author Response · Authors · 2024-08-14
> >
> > > ***The comment remains the same, that something that has explicit optimization with respect to preference intuitively should outperform something that has not been optimized whatsoever. This reviewer still believes the user study is unfair, and that it is completely intuitive and expected that an SD-Turbo + ReNO should outperform non-optimized SD-Turbo with respect to preference, because SD-Turbo+ReNO was explicitly optimized to resepct it! In fact, it is troubling that even with optimization it does not outperform DALL-E 3; intuitively these optimized techniques should always outperform non-optimized ones.***
> >
> > We agree that SD-Turbo + ReNO should outperform SD-Turbo as long as the optimization was done correctly given robust T2I reward models. However, the margin of increase is not clear apriori, see e.g. the performance increase of DOODL in the author rebuttal and Table 3 in the additional PDF. Additionally, outperforming 50-step models that are 2-5x bigger with 10x the compute used for training is not at all to be expected purely from noise optimization.
> >
> >
> > > ***It is strange that the authors are adamant on staying with a one-step diffusion model, refusing to even try some distilled technique that can generate samples in 2 steps or 6 steps (e.g. some LCM models).***
> >
> > Even with a 2-step model, the VRAM requirement will significantly increase. Thus, e.g., 2-step HyperSDXL will not fit into 40GB anymore, making it impractical as a general image generation model. We thank the reviewer for the suggestion and agree that this is an interesting future research direction.

---

> > ### Comment · Reviewer_EAgt · 2024-08-14
> > **Reviewer Response [3]**
> >
> > Surprisingly, the storytelling and motivation outlined in the general comment is more appealing and clear than what was ultimately presented in the paper.  This reviewer is actually on board with the ultimate motivation of "obtain[ing] the best possible/extremely high-quality text-to-image generation using open-source models and tools".  Leading it towards cheaper models and alternatives while considering memory constraints and navigating closed-source models to motivate "test-time optimization methods [as] a compelling alternative to enhance the generation quality of existing multi-step T2I models".  And then going from test-time optimization of large multi-step models to a single one.  Focusing on numerical insights about expensiveness of existing models (e.g. their RAM) would be compelling - a numbers-focused argument would be welcomed.
> >
> > This story reads much better than the one that was initially provided; where the central focus seems to be about T2I noise optimization broadly - but the only thing demonstrated was for one-step T2I models.  And that mismatch is a big source of motivating confusion because the findings for one-step T2I models do not translate necessarily to general source-noise optimization insights for T2I models.  Currently there also seems to be a big focus on preference optimization (through an ensemble), which distracts from the overall goal of simply "improving generation quality of existing T2I models" which everyone can appreciate.
> >
> > If the storytelling were structured like the roadmap provided above from the onset, this reviewer would appreciate the work and its scope much better.  This reviewer actually strongly encourages the authors to write their paper (esp their Introduction, Abstract, etc.) with this motivating story in mind - to avoid any potential confusion and disappointment in what the authors really seek to tackle.  The focus is *not* on source noise optimization because its implications do not extend (currently) to general diffusion models - the source noise optimization is simply a mechanism for improving test-time generation quality.  The one-step diffusion model is *not* used to sidestep multi-step diffusion models (which raises suspicions of the authors avoiding the worthwhile interesting questions, and raises concerns that their insights are not generally useful or applicable across all T2I models), but a choice made for tractability.  This presentation would allow the reviewer to actually appreciate the results rather than be focused on the one-step T2I choice from the get-go and be disappointed at the lack of useful general insights.
> >
> > This reviewer appreciates the proposed story here and can see the pieces fit in - but as the main paper is written currently, the reviewer still feels uncomfortable directly recommending acceptance.  A slight increase of the score will be made out of a common understanding - but still, the reviewer would highly recommend rewriting the motivating portions of the paper (results can obviously remain the same) to fit this proposed storyline.

---

> > > ### Author Response · Authors · 2024-08-14
> > >
> > > Thank you for your insightful feedback. Your comments have helped us recognize areas where we can better articulate our existing work. We plan to refine the presentation in the Abstract, Introduction, and Section 2 to more clearly communicate our motivation and contributions. Specifically, we will emphasize:
> > >
> > > 1. Our primary goal is to maximize the performance of T2I models within significant resource constraints.
> > > 2. Test-time optimization emerges as a promising direction for this goal. However, our experiments with DOODL reveal limitations in achieving the desired balance of quality and efficiency.
> > > 3. Consequently, we focus on one-step models as a tractable starting point, introducing a novel human preference reward model based approach that leverages complementary strengths to boost overall image generation performance.
> > >
> > > For example, we plan to update the following sentence in the Abstract to better reflect this narrative:
> > >
> > > "In this work, we propose Reward-based Noise Optimization (ReNO), a novel approach that enhances T2I models at inference by optimizing the initial noise based on the signal from one or multiple human preference reward models."
> > >
> > > ->
> > >
> > > "In this work, we provide a new perspective on generally improving T2I generation through Reward-based Noise Optimization (ReNO), a novel approach that enhances one-step T2I models at inference. ReNO optimizes the initial noise based on signals from multiple human preference reward models, offering a unique solution to improve generation quality within strict computational constraints."
> > >
> > > > ***Currently there also seems to be a big focus on preference optimization (through an ensemble), which distracts from the overall goal of simply "improving generation quality of existing T2I models" which everyone can appreciate.***
> > >
> > > Additionally, we plan to lower the emphasis on the ensemble of reward models and discuss it as a tool to enhance image generation quality in our updated manuscript.
> > >
> > > > ***The focus is not on source noise optimization because its implications do not extend (currently) to general diffusion models - the source noise optimization is simply a mechanism for improving test-time generation quality. The one-step diffusion model is not used to sidestep multi-step diffusion models (which raises suspicions of the authors avoiding the worthwhile interesting questions, and raises concerns that their insights are not generally useful or applicable across all T2I models), but a choice made for tractability.***
> > >
> > > We agree with your assessment and will more clearly clarify in our revised manuscript that our focus is on improving test-time generation quality, with source noise optimization as a mechanism and one-step models chosen for tractability, rather than to sidestep multi-step models or avoid broader questions in noise optimization for diffusion models.
> > >
> > > We believe this revised framing will more effectively communicate the significance and broader impact of our work in the context of practical T2I model deployment and optimization.

---

### Official Review · Reviewer_mwA2 · 2024-07-11

**Soundness:** 3
**Presentation:** 3
**Contribution:** 2
**Rating:** 4
**Confidence:** 3

**Summary:**

The paper introduces Reward-based Noise Optimization (ReNO), a novel approach to enhance Text-to-Image (T2I) models at inference by optimizing the initial noise based on human preference reward models. ReNO significantly improves model performance within a computational budget of 20-50 seconds, outperforming all current open-source T2I models and being preferred almost twice as often as the popular SDXL model in user studies. Additionally, ReNO-optimized models demonstrate superior efficiency, surpassing widely-used models like SDXL and PixArt-alpha with the same computational resources.

**Strengths:**

1. A new approach that optimizes the initial noise in T2I models at inference time using gradient ascent, which enhances model performance significantly.

2. outperforming all current open-source T2I models and being preferred almost twice as often as the popular SDXL model in user studies. Additionally, ReNO-optimized models demonstrate superior efficiency, surpassing widely-used models like SDXL and PixArt-alpha with the same computational resources.

**Weaknesses:**

Noise optimization is a bit like an "adversarial attack" that achieves its goal by adding noise to the original input, but the only drawback of the method is the time cost, as using gradient descent to obtain noise requires a lot of iterations. Therefore, optimizing time is a crucial factor that requires attention. The paper points out that the optimization time takes 20-50 seconds, and it says that one-step t2i is used. What if multi-step reasoning is directly used? Is the effect better than optimizing noise?

**Questions:**

if the author can significantly reduce the number of optimization iterations, it would be a good method, and the author can refer to some adversarial attack methods. I hope to see the author improve on optimizing time

**Limitations:**

The paper points out that the optimization time takes 20-50 seconds, The optimization time is too long and not easy to use

---

> ### Author Rebuttal · Authors · 2024-08-07
>
> We thank the reviewer for their detailed comments and are especially glad they emphasize the significant enhancement achieved by ReNO. Below, we address the concerns raised in the review.
>
> > ***Therefore, optimizing time is a crucial factor that requires attention. The paper points out that the optimization time takes 20-50 seconds, and it says that one-step t2i is used. What if multi-step reasoning is directly used? Is the effect better than optimizing noise?***
>
> We would like to point out that ReNO actually provides an efficient formulation to enhance T2I models even compared to existing multi-step T2I models. For instance, in Figure 5, we show that ReNO outperforms existing open-source models such as multi-step SDXL and PixArt-$\alpha$ with the same compute budget. Even with 10-15 iterations (4-10 seconds depending on the one-step model), we see significant improvements in prompt following and visual quality, as shown in Figure 5. Additionally, ReNO is much faster than other noise optimization methods, as mentioned in the global rebuttal.
>
> > ***Noise optimization is a bit like an "adversarial attack" that achieves its goal by adding noise to the original input***
>
> While we agree that there are connections to the literature on adversarial attacks, we would like to emphasize that the changes in the generated images are specifically not adversarial. We agree that leveraging different optimization techniques, e.g. from the adversarial attack literature, to reduce the number of iterations/time required for convergence could be an interesting future direction.

---

> > ### Comment · Reviewer_mwA2 · 2024-08-13
> > **Response to rebuttal**
> >
> > Thank you for the responses.
> >
> > Referring to the opinions of other reviewers, I think the proposed method is not universal and time-consuming, so I keep my score.

---

> > > ### Author Response · Authors · 2024-08-13
> > >
> > > The reviewer mentions that our method is not universal and time-consuming, and points to the other reviews. It would be much appreciated if the reviewer could provide more details, since in our rebuttal to each of the other reviews, we have thoroughly clarified these points.
> > >
> > > > ***Universality of ReNO:***
> > >
> > > In this work, we are tackling the most general form of Text-to-Image generation. ReNO-enhanced one-step models consistently surpass the performance of all current open-source Text-to-Image models across a variety of **general** T2I benchmarks and a comprehensive user study.
> > >
> > > > ***Time-consuming:***
> > >
> > > We address this point in the rebuttal above. ReNO enhanced SD-Turbo outperforms existing open-source models such as multi-step SDXL and PixArt-$\alpha$ **with the same compute budget**. This shows that ReNO is not time-consuming but actually time-efficient.

---

### Official Review · Reviewer_8y7L · 2024-07-12

**Soundness:** 3
**Presentation:** 4
**Contribution:** 4
**Rating:** 6
**Confidence:** 5

**Summary:**

The paper presents a novel approach called Reward-based Noise Optimization (ReNO) to enhance the performance of one-step Text-to-Image (T2I) models. ReNO optimizes the initial noise of T2I models using a human preference reward model, addressing the limitations of current T2I models in capturing complex details in compositional prompts. The method shows promising results across four different one-step models on T2I-CompBench and GenEval benchmarks, outperforming open-source T2I models and achieving comparable performance to a proprietary model. ReNO is computationally efficient and improves the quality of generated images, as demonstrated through user studies.

**Strengths:**

1. It is reasonable to introduce a distilled one-step T2I model to address the notorious issue of exploding/vanishing gradients that exist in T2I diffusion models.
2. ReNO improves the accuracy of T2I models in capturing intricate details in complex prompts. It consistently surpasses the performance of popular open-source T2I models.
3. ReNO is applicable to existing models, avoiding the need for retraining from scratch. The approach is based on human preference, enhancing the alignment with desired outputs.

**Weaknesses:**

1. ReNO is essentially a runtime optimization approach that leverages advanced reward models to achieve state-of-the-art text-to-image generation capabilities. Indeed, the authors opt to optimize initial noise as a learnable parameter. Could I choose to make the parameters of a UNet learnable to achieve a similar objective?
2. Have the authors attempted to apply ReNO to text-to-image diffusion models, such as SD, using techniques like gradient checkpoint and LoRA?
3. Has the proposed ReNO been tested on video generation diffusion models for text-to-video tasks?
4. The authors are encouraged to include the following references:
   - Guided image synthesis via initial image editing in diffusion model, ACM MM 2023
   - InitNO: Boosting Text-to-Image Diffusion Models via Initial Noise Optimization, CVPR 2024

**Questions:**

Overall, ReNO presents a simple yet effective solution to enhance text-to-image diffusion models without the need for additional training. Generally, I have a positive outlook. Please refer to the Weaknesses section for a detailed list of questions and suggestions.

**Limitations:**

The authors provide Limitations and Broader Impact.

---

> ### Author Rebuttal · Authors · 2024-08-07
>
> We thank the reviewer for their detailed comments. We are especially glad that they appreciated the simplicity and effectiveness of ReNO. Below, we address the concerns raised in the review.
>
> > ***ReNO is essentially a runtime optimization approach that leverages advanced reward models to achieve state-of-the-art text-to-image generation capabilities. Indeed, the authors opt to optimize initial noise as a learnable parameter. Could I choose to make the parameters of a UNet learnable to achieve a similar objective? Have the authors attempted to apply ReNO to text-to-image diffusion models, such as SD, using techniques like gradient checkpoint and LoRA?***
>
> Yes, we also briefly tried optimizing the parameters of the U-Net with LoRA. This leads to the model generating images with visual artifacts which is caused due to "reward-hacking". In contrast, only optimizing the noise keeps the model untouched and, thus, does not lead to it generating adversarial images as long as the noise stays in distribution. Moreover, fine-tuning with reward models poses a number of other challenges. [lines 40-51] For example, the SDXL U-Net has 2.6B parameters, which makes it computationally infeasible to fine-tune at inference for every prompt. Even with LoRA and gradient checkpointing (as done by AlignProp[57]), this would still have 10M+  parameters to train (vs ~16k parameters for the noise optimization), which would take several minutes, if not hours (AlignProp fine-tunes SD1.5 on 4 GPUs for 24 hours).
>
> > ***Has the proposed ReNO been tested on video generation diffusion models for text-to-video tasks?***
>
> Our noise optimization framework is directly applicable to video generation. However, at the moment, while there are human preference reward models for video generation, there are no one-step video generation models publicly available, which are otherwise prohibitively expensive to train in a resource-constrained setting. Once open-source one-step video models are available, ReNO would be ideally suited to further enhance the performance of video generation.
>
> > ***The authors are encouraged to include the following references***
>
> We thank the reviewer for pointing out these related works, especially the related concurrent InitNO, and we will add these to the paper. While InitNO also looks to optimize the initial noise, this is done by computing a loss function using attention maps as opposed to optimization with human preference reward objectives.

---

### Official Review · Reviewer_UuFQ · 2024-07-13

**Soundness:** 3
**Presentation:** 3
**Contribution:** 3
**Rating:** 7
**Confidence:** 4

**Summary:**

The paper introduces Reward-based Noise Optimization (ReNO), a novel method to improve Text-to-Image (T2I) models by optimizing the initial noise during inference using human preference signals. This approach addresses the limitations of current fine-tuning methods, which often lead to "reward hacking" and poor generalization. By utilizing one-step diffusion models, ReNO enhances image quality and adherence to complex prompts without retraining, achieving significant performance improvements on benchmarks like T2I-CompBench and GenEval. Extensive user studies demonstrate that ReNO models are preferred nearly twice as often as popular models like SDXL, showcasing their efficiency and effectiveness in enhancing T2I model performance and user satisfaction.

**Strengths:**

1. Optimizing the initial noise input during inference to improve image quality and prompt fidelity, which is an innovative angle compared to typical model fine-tuning approaches.
2. Conducts extensive experiments on multiple challenging benchmarks (T2I-CompBench, GenEval, Parti-Prompts) to evaluate the method.
3. Compares against a wide range of baselines and state-of-the-art models, including proprietary ones like DALL-E 3 and Stable Diffusion. 4. Analyzes the impact of different reward models and optimization iterations.
5. Clearly explains the motivation and approach of ReNO.
6. Demonstrates competitive performance with proprietary models like SD3, despite using smaller open-source models as a base.
7. Provides a practical method to enhance text-to-image models at inference time with reasonable computational cost (20-50 seconds per image).

**Weaknesses:**

1. Limited analysis of potential negative impacts or failure modes:

The paper does not thoroughly discuss potential downsides or risks of their approach. For example:

- Could optimizing for reward models lead to unexpected or undesirable outputs in some cases?
- Are there risks of amplifying biases present in the reward models?
- Could this approach be misused to generate more convincing deepfakes or misleading images?

2. Limited comparison to related optimization approaches:

The paper compares to some baseline models, but doesn't thoroughly compare to other test-time optimization methods for text-to-image models. Comparisons to approaches like:

- DOODL (Kerras et al., 2022)
- D-Flow (Ben-Hamu et al., 2023)

Would help contextualize the novelty and advantages of ReNO.

3. Insufficient analysis of impact on image diversity:
- The paper doesn't thoroughly examine whether optimizing for rewards reduces the diversity of generated images. Some analysis of how ReNO affects the distribution of outputs would be valuable. (Specially theoretical)

Aside from the mentioned points, everything else was satisfactory, and I enjoyed the paper!

**Questions:**

1. Reward model robustness:
- How sensitive is ReNO to the choice of reward models? Have you observed any cases where optimizing for certain reward models leads to unexpected or undesirable results? This could help understand the robustness and potential limitations of the approach.

2. Computational efficiency:
- Could you provide more details on how ReNO's performance scales with the number of optimization steps and computational budget? Is there a clear point of diminishing returns?

3. Diversity of outputs:
- Does optimizing for reward models potentially reduce the diversity of generated images? Have you conducted any analysis on how ReNO affects the distribution of outputs compared to the base models?

4. Integration with other techniques:
- How might ReNO complement or interact with other techniques for improving text-to-image models, such as fine-tuning or prompt engineering? (Out of curiosity)

**Limitations:**

The authors acknowledge some limitations of their approach, particularly in the "Limitations" section. They mention:

- Convergence of different models to similar performance levels, potentially due to limitations in reward models.
- The increased VRAM requirements of their method.
- Persistent challenges in generating humans, rendering text, and modeling complex compositional relations.
- They briefly mention the possibility of hallucination in their method, which is a relevant concern for AI-generated content.


I think they addressed the limitations adequately.

---

> ### Author Rebuttal · Authors · 2024-08-07
>
> We thank the reviewer for their detailed comments. We are especially glad that they enjoyed the paper. Below, we address the concerns raised in the review.
>
> >  ***Limited comparison to related optimization approaches***
>
> We address the comparison to DOODL in the global response. For D-Flow, the proposed method takes even longer (30-40 minutes for a 350M parameter model).
>
> > ***Insufficient analysis of impact on image diversity / Diversity of outputs***
>
> We address this in the global response. Would you like us to analyze any other metrics to measure the distribution of generated images with ReNO compared to without ReNO?
>
> > ***Limited analysis of potential negative impacts or failure modes***
>
> ReNO can successfully enhance the general capabilities of existing T2I models, including better prompt following and visual quality. As with all T2I models, this can be used for positive and negative applications. Additionally, since the objective is to optimize for the human preference reward models, the model is biased towards aspects that these models especially focus on. Reward models could also be prone to biases because of, e.g., their training data and, thus, this could be a potential risk in amplifying biases in existing models and a source of undesirable outputs. However, ReNO also opens up the possibility of including a bias-mitigating reward model.
>
> > ***Reward model robustness***
>
> We would like to highlight Tables 1 & 7, where we benchmark the choice of different reward models. We did observe undesirable outputs **without** the noise regularization, as sometimes images with severe artifacts are generated, because of the noise going out of distribution, that still achieve a high reward score. Additionally, the reward models focus on some aspects more (e.g., colors, counting) than others (e.g., spatial understanding in the prompt). Specifically, we found that for a prompt including "under/above" or "left/right" the reward model scores can be very similar independent of which one is chosen for the same image.
>
> > ***Computational Efficiency***
>
> We would like to highlight Figure 5, where we show the improvement in attribute binding on T2I-Compbench with increasing iterations. We see rapid improvements for the first 10-15 iterations (4-6 seconds for SD-Turbo on one A100), where it already surpasses popular open-source multi-step models like multi-step SDXL and PixArt-$\alpha$. We support this with some qualitative examples in Figure 6. After 50 iterations (20 seconds), we see diminishing returns and see no major gains beyond 75 iterations (30 seconds).
>
> > ***Integration with other Techniques:***
>
> Our experiments (Tables 2 & 3) indicate that we get better results with stronger base models. Therefore, fine-tuning base models to improve their performance or orthogonal techniques like prompt engineering should enhance the results further and be straightforward to incorporate together with ReNO.

---

> > ### Comment · Reviewer_UuFQ · 2024-08-14
> >
> > Thank you for addressing my concern. I think the paper is now is a good shape (especially after the new experiments). I'd like to maintain my score.

---

### Author Rebuttal · Authors · 2024-08-07

We would like to thank all reviewers for their time and their detailed and insightful comments. We appreciate their recognition of the ReNO's significant enhancement in general performance (*UuFQ*,*8y7L*,*mwA2*), and its novelty (*UuFQ*, *mwA2*), clarity (*UuFQ*, *8y7L*), and practicality (*UuFQ*, *8y7L*). We want to highlight the new experiments we conducted and some of the joint questions here while additionally we address each raised issue with individual replies for each reviewer. We repeat the major contributions of ReNO here:

- Practical Noise Optimization for T2I Models: ReNO demonstrates that one-step models can make noise optimization a practical tool for generally enhancing Text-to-Image generation. It significantly improves efficiency in both time (20-50 seconds vs 20 minutes per image for previous noise optimization methods) and memory usage. This approach makes optimizing for human preference reward models during inference practically possible, substantially enhancing both prompt adherence and visual quality.
- Significantly Enhanced Performance: ReNO-enhanced one-step models achieve results that are on par with much larger, better-trained closed-source models while outperforming popular open-source models by large margins on standard benchmarks. Notably, given the same computational budget, one-step models with ReNO outperform widely used multi-step models, offering superior results with enhanced efficiency. This demonstrates the effectiveness of aligning T2I outputs with human preferences during inference, even with the constraints of a one-step model, and elucidates the power of noise optimization in a novel more general setting.

---

**Diversity evaluation.** We've conducted a diversity analysis in response to the reviewers' questions about ReNO's impact on diversity. We generated images with 50 different seeds for 10 prompts from each of the 11 challenges of PartiPrompts, totaling 110 prompts. Then, for each prompt, we evaluate the diversity over 50 seeds by computing the mean pairwise LPIPS and DINO score. The higher these two scores are, the less diverse the generated images across seeds. We report the mean and standard deviation across all prompts in the following table, as well as in the additional pdf.

|                   | LPIPS                | DINO                 |
| ----------------- | -------------------- | -------------------- |
| SD-Turbo          | 0.382 ± 0.043        | 0.770 ± 0.101        |
| SD-Turbo + ReNO   | *0.246* ± 0.046 | *0.712* ± 0.132 |
| SD2.1 (50-step)   | **0.243** ± 0.049    | **0.623** ± 0.150    |
| SDXL-Turbo        | 0.391 ± 0.044        | 0.835 ± 0.073        |
| SDXL-Turbo + ReNO | **0.291** ± 0.041    | *0.763* ± 0.116 |
| SDXL (50-step)    | *0.351* ± 0.042 | **0.700** ± 0.128    |

Remarkably, ReNO actually significantly increases the diversity of one-step models. As we believe this is a significant finding, we plan to include this in the main text of the paper. We also provide some non-cherry-picked results for the first 5 seeds in the pdf.

We hypothesize that the reason for this increased diversity is that ReNO optimizes the noise away from the zero mean of the noise distribution, thus creating more diverse noises compared to sampling from the standard Gaussian. Even though we regularize the noise to stay in distribution, we do not enforce this. To validate this hypothesis, we compute the standard deviation across all noises before and after ReNO-optimization. As expected, the standard deviation for the initial noise across this sample size (110 prompts * 50 seeds) is *1.0000*. In contrast, the standard deviation of ReNO-optimized noise is *1.0039*, which confirms this hypothesis.

---

**Comparison to DOODL.** Due to the fact that DOODL takes more than 80 days on one A100 to evaluate on T2I-Compbench, we were unable to make comparisons to it on the standard  benchmarks. The following table illustrates this point:

|                 | sec/iter (total) | T2I-CompBench duration | VRAM | \#params |
| --------------- | ---------------- | ---------------------- | ---- | -------- |
| SD2.1 + DOODL (CLIP)   | 24s (20min)      | 83.33 A100 days        | 40GB | 860M     |
| SD-Turbo + ReNO (only CLIP) | 0.2s (10s)       | 0.63 A100 days         | 10GB | 860M  |
| SD-Turbo + ReNO | 0.4s (20s)       | 1.25 A100 days         | 15GB | 860M  |

Furthermore, when employing a larger model such as 50-step SDXL or multiple reward models, DOODL's VRAM requirement exceeds 40GB, making it infeasible even on A100 GPUs. Additionally, since DOODL optimizes multi-step models, it encounters problems with exploding gradients, which may compromise its noise optimization efficiency compared to ReNO, despite its extended runtime. To substantiate this, we evaluated DOODL on the first 50 prompts from T2I-CompBench's three attribute binding tasks. Our analysis includes both VQA evaluation results and changes in the optimized CLIPScore, effectively measuring the efficacy of the 50-step optimization process. We compare SD2.1 + DOODL (using CLIPScore) against SD-Turbo + ReNO (using CLIPScore) and ReNO with all considered reward models.

|Model|Color ↑|Shape ↑|Texture ↑|CLIPScore ↑|
|---|---|---|---|---|
|SD2.1|33.4|52.4|63.4|0.261|
|SD2.1 + DOODL (CLIP)|38.5 (+5.1)|51.6 (-0.8)|64.6 (+1.2)|0.289 (+0.03)|
|SD-Turbo|60.4|48.5|61.8|0.362|
|SD-Turbo + ReNO (only CLIP)|70.1 (*+9.7*)|66.9 (*+18.4*)|79.6 (*+18.2*)|0.483 (**+0.12**)|
|SD-Turbo + ReNO (all)|82.1 (**+21.7**)|77.4 (**+28.9**)|82.8 (**+21.0**)|0.437 (*+0.08*)|

We observe that ReNO achieves substantially higher gains compared to DOODL, both w.r.t. the CLIP loss that we optimize for and also the independent VQA evaluation. Note that the results from DOODL are in line with those reported in their paper, where they report increases in CLIPScore by 0.026 and 0.031. To provide a more comprehensive comparison between DOODL and ReNO, we plan to include this analysis and its results in the paper's Appendix.

---

### Author Response · Authors · 2024-08-11

Dear Reviewers,

Thanks for your efforts in reviewing our paper. Specifically, we want to thank you for asking important questions, which led us to study the change in diversity through ReNO and provide a more thorough comparison to DOODL. Please let us know if our response addresses all your concerns or if you would like us to provide any other clarifications.

---

### Author Response · Authors · 2024-08-14
**End of Discussion Period**

Dear Reviewers,

As the discussion period draws to a close, we would like to express our sincere gratitude for your initial review of our paper. We appreciate the recognition of our work's contributions in terms of performance enhancement, novelty, clarity, and practicality.

We hope that in our rebuttal, we have successfully addressed each point raised and provided valuable additional analysis based on your feedback. This includes a diversity study of ReNO, where we found that ReNO significantly increases diversity, and a more comprehensive comparison with DOODL (ReNO is 60x faster with 4x better metric optimization). Additionally, we want to further highlight that ReNO-enhanced one-step models outperform multi-step models with the same computational budget by summarizing Figure 5 in the following table, illustrating the efficacy of our proposed method:

| Model                      | T2I-CompBench Average | Time    |
| -------------------------- | --------------------- | ------- |
| SDXL                       | 0.5804                | ~7 sec  |
| SD-Turbo                   | 0.5217                | 0.4 sec |
| SD-Turbo + ReNO (10 steps) | 0.6315                | 4 sec   |
| SD-Turbo + ReNO (15 steps) | 0.6497                | 6 sec   |
| SD-Turbo + ReNO (20 steps) | 0.6653                | 8 sec   |
| SD-Turbo + ReNO (25 steps) | 0.6804                | 10 sec  |
| SD-Turbo + ReNO (30 steps) | 0.6966                | 12 sec  |
| SD-Turbo + ReNO (40 steps) | 0.7082                | 16 sec  |
| SD-Turbo + ReNO (50 steps) | 0.7180                | 20 sec  |

We believe these clarifications and additions further demonstrate the strengths of our approach.

We would greatly appreciate your consideration of our rebuttal as you formulate your final evaluations. We again want to thank the reviewers for their time and expertise throughout the review process. The reviewers' suggestions have, without a doubt, improved the breadth and quality of our submission.

Kind regards,

Anonymous Authors

---

### Decision · Program_Chairs · 2024-09-25

**Decision:**

Accept (poster)

**Comment:**

**Summary:**

The paper introduces Reward-based Noise Optimization (ReNO), a method to improve Text-to-Image (T2I) models by optimizing initial noise during inference using human preferences. ReNO enhances image quality and prompt fidelity without retraining, showing significant improvements on benchmarks and user preference over models like SDXL.

**Strengths:**

- Novel approach of optimizing noise during inference for better image quality.
- Extensive experiments and comparisons with state-of-the-art models.
- Practical method with clear motivation and reasonable computational cost.

**Weaknesses:**

- Limited discussion on risks, such as bias amplification or unintended outputs. Insufficient analysis of ReNO’s impact on image diversity.
- Incomplete comparisons with other test-time optimization methods.

**Reason to Accept:**

The paper offers an innovative and effective method for improving T2I models, supported by strong experimental results, making it a valuable contribution despite some areas needing further analysis.